# PARALLEL $Q$-LEARNING: SCALING OFF-POLICY REINFORCEMENT LEARNING

## ABSTRACT

Reinforcement learning algorithms require a long time to learn policies on complex tasks due to the need for a large amount of training data. With the recent advances in GPU-based simulation, such as Isaac Gym, data collection has been sped up thousands of times on a commodity GPU. Most prior works have used on-policy methods such as PPO to train policies in Isaac Gym due to its simpleness and effectiveness in scaling up. Off-policy methods are usually more sample-efficient but more challenging to be scaled up, resulting in a much longer wall-clock training time in practice. In this work, we presented a novel Parallel $Q$-Learning (PQL) framework that is substantially faster in wall-clock time and achieves better sample efficiency than PPO. Our key insight is to parallelize the data collection, policy function learning, and value function learning as much as possible. Different from prior works on distributed off-policy learning, such as Apex, our framework is designed specifically for massively parallel GPU-based simulation and optimized to work on a single workstation. We demonstrate the capability of scaling up $Q$ learning methods to tens of thousands of parallel environments. We also investigate various factors that can affect policy learning speed, including the number of parallel environments, exploration schemes, batch size, GPU models, etc.

## 1 INTRODUCTION

Reinforcement learning (RL) has achieved impressive results on many problems: video games (Berner et al., 2019; Mnih et al., 2015), robotics (Kober et al., 2013; Miki et al., 2022), drug discovery (Popova et al., 2018) and others. A primary challenge in using RL is the need for large amounts of training data. One way to tackle this problem is by improving off-policy RL algorithms (Mnih et al., 2015; Lillicrap et al., 2015) that make better use of data than on-policy algorithms (Schulman et al., 2017; Mnih et al., 2016). Another strategy is to substantially reduce the need for real-world data collection by training in simulation. Recent works have achieved remarkable success in deploying policies trained in simulation to the real-world (Hwangbo et al., 2019; OpenAI et al., 2020; Margolis et al., 2022; Miki et al., 2022). In a sim-to-real training paradigm, it is not the amount of training data but the wall-clock time that is a major constraint. Faster training, measured as wall-clock time, speeds up the experimentation cycle and unlocks the potential for addressing a broader range of more complex problems that currently take a long time to train.

The need for faster training has been recognized in the literature resulting in several distributed learning frameworks (Horgan et al., 2018; Espeholt et al., 2018). Typically these frameworks operate at the server scale requiring hundreds or thousands of computers, making them infeasible for most researchers. Many of these computers are used to run multiple copies of a "slow" simulator in parallel to feed the learning process. Recent advances in GPU-based simulation, such as Isaac Gym (Makoviychuk et al., 2021), have mitigated the need for a large number of machines for parallel simulation by enabling the parallel simulation of thousands of environments on a commodity GPU on a single workstation. One natural question to ask is, what RL algorithm is apt in such a setting? Many prior works (Allshire et al., 2021; Rudin et al., 2022; Chen et al., 2022) use PPO (Schulman et al., 2017) for training agents in Isaac Gym due to its simplicity and easy-to-scale nature.

Intuitively, by virtue of requiring lesser data than on-policy algorithms, off-policy algorithms should reduce the wall-clock time of training. However, a naive implementation of off-policy algorithms

such as DDPG (Lillicrap et al., 2015) or SAC (Haarnoja et al., 2018) without making use of parallel environments usually requires a much longer training time than PPO, even though these algorithms are more sample efficient. We build upon distributed frameworks for Q-learning developed and deployed in server-scale settings (Horgan et al., 2018; Nair et al., 2015) to leverage GPU-based simulation on a single workstation. We present an approach to scale up $Q$-learning, **P**arallel **Q**-**L**earning (**PQL**), that can be deployed on a workstation to leverage thousands of environment simulations running in parallel efficiently. The key factor that boosts the learning speed in PQL is that we parallelize the data collection, policy function update, and value function update as much as possible on a single workstation. Such parallelization would be non-trivial for on-policy algorithms such as PPO as the policy update require on-policy interaction data, which means the data collection and the policy update have to happen in sequence. **PQL** outperforms state-of-the-art algorithms such as PPO in terms of both wall-clock and data efficiency.

We empirically investigate the effectiveness of our method on six Isaac Gym tasks (Makoviychuk et al., 2021), demonstrating the superior performance of **PQL** against commonly used state-of-the-art (SOTA) RL algorithms. We also analyze several important factors that can affect learning speed, such as the number of parallel environments, batch size, balancing between parallel processes performing simulation, data-collection and learning, exploration scheme, replay buffer size, number of GPUs, different GPU hardware, etc. Other noteworthy findings are: (i) we empirically found that DDPG performs better than SAC when using a large number of parallel environments. (ii) We can mitigate the need for tuning the hyper-parameter controlling exploration. Overall, while previous distributed learning frameworks were accessible only to researchers with access to server-scale compute, we hope our framework leveraging recent advances in GPU-based simulation will be a useful tool for the broader research community training RL agents on a single workstation.

## 2 RELATED WORK

**Massively Parallel Simulation**   Simulation has been an important tool for various research fields, including robotics, drug discovery, physics, etc. In the past, researchers have used simulators like MuJoCo (Todorov et al., 2012), PyBullet (Coumans & Bai, 2016) for rigid body simulation. Recently, there has been a new wave of development in GPU-based simulation, such as Isaac Gym (Makoviychuk et al., 2021). GPU-based simulation has substantially improved the simulation speed, allowing a massive amount of parallel simulation on a single commodity GPU. It has been used in various challenging robotics control problems such as quadruped locomotion (Rudin et al., 2022; Margolis et al., 2022), dexterous manipulation (Chen et al., 2022; Allshire et al., 2021). With the advent of fast simulation, one can get much more environment interaction data in the same training time as before. This poses a challenge to the reinforcement learning algorithm in making the best use of the massive amount of data. A straightforward way is to use on-policy algorithms such as PPO, which can be scaled up easily and is also the default algorithm that researchers use in Isaac Gym. However, on-policy algorithms are less data efficient. In our work, we investigate how to scale up off-policy algorithms to achieve both higher sample efficiency as well as shorter wall-clock training time for massively parallel simulation. Our parallel training framework works on a commodity workstation without requiring a big compute cluster.

**Distributed Reinforcement Learning**   Model-free reinforcement learning typically requires a large number of environment interactions (low sample efficiency). One way to speed up policy learning is by distributed training. There have been numerous works developing different distributed reinforcement learning frameworks. One line of work focuses on $Q$-learning methods. Gorila (Nair et al., 2015) distributes DQN agents to many machines where each machine has its local environment, replay buffer, value learning, and uses asynchronous SGD for a centralized $Q$ function learning. Similarly, Popov et al. (2017) applies asynchronous SGD to the DDPG algorithm (Lillicrap et al., 2015). Combining with prioritized replay (Schaul et al., 2015), $n$-step returns (Sutton, 1988), and double-Q learning (Hasselt, 2010), Horgan et al. (2018) (Ape-X) parallelizes the actor thread (environment interactions) for data collection and uses a centralized learner thread for policy and value function learning. Built upon Ape-X, Kapturowski et al. (2018) adapts the distributed prioritized experience replay for RNN-based DQN agents.

Another line of work improves the training speed on policy gradient methods. A3C (Mnih et al., 2016) uses asynchronous SGD across many CPU cores, with each one of them running an actor

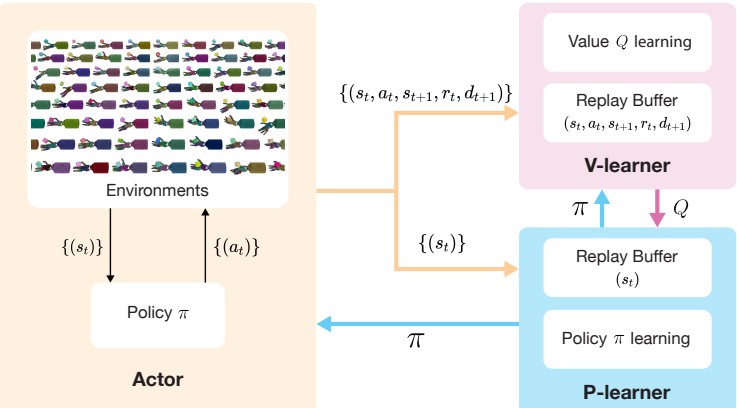

Figure 1: Overview of Parallel $Q$ Learning (PQL). We have three concurrent processes running: **Actor**, **P-learner**, **V-learner**. **Actor** collects interaction data. **P-learner** updates the policy network. **V-learner** updates the $Q$ functions.

learner on a single machine. Babaeizadeh et al. (2016) develops a hybrid CPU/GPU implementation of A3C, but it can have poor convergence due to the stale off-policy data being used for the on-policy update. Espeholt et al. (2018) (IMPALA) introduces an off-policy correction scheme (V-trace) to mitigate the lagging issue between the actors and learners in distributed on-policy settings. Espeholt et al. (2019) further improves the IMPALA training speed by moving the policy inference from the actor to the learner. Clemente et al. (2017) parallelizes the environments for synchronous advantage actor-critic. Heess et al. (2017) proposes a distributed version of PPO (Schulman et al., 2017) for training various locomotion skills in a diverse set of environments. Wijmans et al. (2019) develops a decentralized version of distributed PPO to mitigate the synchronization overhead between different actor processes and applies it to a point-goal navigation task.

Our framework is closest related to Ape-X (Horgan et al., 2018), but has a number of key differences. First, our framework is specifically designed for massively parallel GPU-based simulation. Our framework is optimized for a single-machine setup, which can further democratize large-scale RL research. Second, we put a replay buffer on the same process as the learners (policy function learning and $Q$ function learning). This can reduce the communication cost between the replay buffer and the learners. Third, working with a single machine presents new challenges in balancing the compute resource between different parallel processes. Our framework offers a mechanism to balance the compute resource among different processes.

## 3 METHOD

We developed a parallel off-policy training framework, **P**arallel **Q**-**L**earning (**PQL**), for massively-parallel GPU-based simulation. Our intuition is simple: parallelize the computation as much as possible on a workstation. A typical actor-critic $Q$-learning method involves three components: a policy function, a $Q$ value function, and an environment. In many popular reinforcement learning libraries (Raffin et al., 2021; Stooke & Abbeel, 2019; Weng et al., 2021), these three components run sequentially. A typical workflow is as follows: agents roll out the policy in the environments and collect interaction data which will be added to a replay buffer, then the value function will be updated to minimize the Bellman error, after which the policy function will be updated to maximize the $Q$ values. In terms of the training wall-clock time, such a sequential scheme slows down the training as each component needs to wait for the other two to finish to proceed. To maximize the learning speed and reduce the waiting time, our idea is to parallelize the computation of all three components. On the other hand, parallizing the data collection and policy/value function update would be non-trivial for on-policy methods like PPO as the policy update requires on-policy interaction data.

Our framework is optimized for training speed in terms of wall-clock time and can be readily applied on a workstation. Our framework is built upon DDPG (Lillicrap et al., 2015), but can be easily extended to other off-policy algorithms such as SAC (Haarnoja et al., 2018). To reduce the overestimation issue in $Q$ values, we also add double $Q$ learning (Hasselt, 2010). To improve the learning efficiency, we also use $n$-step returns (Sutton, 1988) for $Q$ function learning. We will use the following notations in the following paper: $s_t$ for observation data at time step $t$, $a_t$ for action command at time step $t$, $r_t$ for the reward at time step $t$, $d_t$ for whether the environment terminates

at time step $t$, $\pi(s_t)$ for the policy network, $Q(s_t, a_t)$ for the $Q$ network, $Q'(s_t, a_t)$ for the target $Q$ network, $N$ for the number of parallel environments.

## 3.1 Framework Overview

Our framework consists of three parallel processes:

- **Actor**: collecting interaction data in different environments. We use Isaac Gym (Makoviychuk et al., 2021) as the simulation engine which supports massively parallel environment simulation. In one actor process, we can parallelize tens of thousands of environments on a commodity GPU. The actor process has a local policy network $\pi^a(s_t)$, which is periodically updated with the policy network $\pi^p(s_t)$ in **P-learner**.

- **V-learner**: We create a dedicated process for training the critic functions in DDPG. **V-learner** has a local replay buffer consisting of the transition data $\{(s_t, a_t, s_{t+1}, r_t, d_{t+1})\}$. We put the replay buffer on GPU unless otherwise specified. It also contains one local actor network $\pi^v(s_t)$ which gets updated to $\pi^p(s_t)$ in **P-learner**, two critic networks $Q_1^v(s_t, a_t), Q_2^v(s_t, a_t)$ and two target critic networks $Q_1^{v\prime}(s_t, a_t), Q_2^{v\prime}(s_t, a_t)$. The $Q$ functions are optimized to minimize the mean squared Bellman error. Having a dedicated process for updating the $Q$ functions allows the critic functions to be updated continuously without being interrupted by data collection or actor network updates.

- **P-learner**: We use another dedicated process for updating the actor network $\pi^p(s_t)$. It has a replay buffer of $\{(s_t)\}$, and a local value function $Q^p(s_t, a_t)$ which is periodically updated with $Q_1^v(s_t, a_t)$ in **V-learner**. The actor network $\pi^p(s_t)$ is optimized to maximize the $Q^p(s_t, \pi^p(s_t))$.

Parallelizing each part (data collection, actor, and critic update) of the policy learning allows for more data being collected and more network updates, which can improve the training speed, as we will show in the experiments. We use Ray (Moritz et al., 2017) to achieve parallelization. The pseudo-code for the framework is shown in Algorithm 1, 2, and 3.

**Data Transfer** Suppose there are $N$ parallel environments in the **Actor** process. At each rollout step, the **Actor** rollouts the policy $\pi_a(s_t)$ and generates $N$ pairs of $(s_t, a_t, s_{t+1}, r_t, d_{t+1})$. Then the **Actor** sends the entire batch of interaction data $\{(s_t, a_t, s_{t+1}, r_t, d_{t+1})\}$ to the **V-learner**. Since policy update in **P-learner** only needs state information, **Actor** only needs to send $\{(s_t)\}$ to the **P-learner**.

**Network Transfer** The **V-learner** periodically sends the parameters of the $Q_1^v(s_t, a_t)$ to **P-learner**, which updates the local $Q^p(s_t, a_t)$ in **P-learner**. The **P-learner** sends the actor network $\pi^p(s_t)$ to both the **Actor** and **V-learner**.

Both the data transfer and network transfer happen concurrently.

## 3.2 Balance between Actor, P-learner, and V-learner

Our framework allows the **Actor**, **P-learner**, **V-learner** to run concurrently. However, the data collection frequency, policy network, and $Q$ network update need to be constrained properly. In other words, each process should not be running on its own as fast as possible. For example, prior works (Fujimoto et al., 2018) have shown that updating the policy network less frequently than the $Q$ functions leads to better learning. On the other hand, leaving each process running freely creates more variance in the training speed and learning performance as the simulation speed and network training speed are heavily dependent on the task complexity, network size, hardware, etc. For example, simulation for some tasks might be slower than others. Some tasks might require a deeper policy network or $Q$ networks. Even the GPUs on a machine might have different running conditions at different times, leading to different speeds across processes and further leading to different learning performance. To reduce the variance of our framework's performance, we add explicit control on the ratio among the data collection frequency, policy update frequency, and the $Q$ function update frequency. We define two ratios as follows:

$$\beta_{a:v} := \frac{f_a}{f_v} \qquad \beta_{p:v} := \frac{f_p}{f_v}$$

Figure 2: We experiment on six Isaac Gym tasks: (from left to right) *Ant*, *Humanoid*, *ANYmal*, *Shadow Hand*, *Allegro Hand*, *Franka Cube Stacking*.

where $f_a$ is the number of rollout steps per environment in **Actor** per unit time, $f_v$ is the number of $Q$ function updates in **V-learner** per unit time, $f_p$ is the number of policy updates in **P-learner** per unit time. $\beta_{a:v}$ determines how many $Q$ function updates are performed in the **V-learner** when **Actor** rollouts the policy for one step with $N$ environments. $\beta_{p:v}$ decides how many $Q$ function updates are performed in **V-learner** when **P-learner** updates the policy once. With the ratios being set, we keep monitoring the progress of each process and dynamically adjust **Actor** and **P-learner** speed by letting the process wait if necessary. Controlling the three processes via $\beta_{a:v}, \beta_{p:v}$ brings another advantage on resource allocation. When working with limited compute resources on a single workstation, having the ability to let some of the processes wait allows other processes to use the GPU resource more. For example, if there is only one GPU, and if we put all three processes on the GPU, simulation with a large number of environments can cause very high GPU utilization, slowing down the **P-learner** and **V-learner**. By controlling $\beta_{a:v}, \beta_{p:v}$, the processes can be balanced in terms of how much computing can be used.

## 3.3 MIXED EXPLORATION

Balancing the exploration and exploitation requires extensive hyper-parameter tuning in reinforcement learning. In DDPG, since the policy network outputs the actions deterministically, one needs to add exploration noise explicitly to the action commands during training. The amount of noise being added needs tuning for each task. One common practice to add exploration noise in DDPG is to add uncorrelated and zero-mean Gaussian noise: $a_t = \max(\min(\pi(s_t) + \mathcal{N}(0, \sigma), a_u), a_l)$ (Achiam, 2018; Fujita et al., 2021) where $a_t \in [a_l, a_u]$, and $\sigma$ is the standard deviation of the Gaussian distribution and controls how much noise is being added. One typically needs to tune $\sigma$ to get the best performance for each task. Since different $\sigma$ leads to varied exploration and exploitation balance, given that we can massively parallelize the environments, one natural idea is to use different exploration strategies in different environments, i.e., set different $\sigma$ values for different environments, which we call **mixed exploration**. Similar ideas have been used in some prior works (Horgan et al., 2018; Mnih et al., 2016). In our work, we uniformly generate the noise levels in the range of $[\sigma_{\min}, \sigma_{\max}]$. In other words, for $i^{th}$ environment out of $N$ environments, $\sigma_i = \sigma_{\min} + \frac{i-1}{N-1}(\sigma_{\max} - \sigma_{\min})$ where $i \in \{1, 2, ..., N\}$. In our experiments, we use $\sigma_{\min} = 0.05, \sigma_{\max} = 0.8$ for all the tasks.

## 4 EXPERIMENTS

We evaluate our method on six Isaac Gym benchmark tasks (Makoviychuk et al., 2021). All our experiments are carried out on a single workstation with a few GPUs. We consider the following baselines: (1) **PPO** (Schulman et al., 2017), which is the defacto algorithm used by many prior works (Makoviychuk et al., 2021; Chen et al., 2022; Allshire et al., 2021) that use Isaac Gym for simulation, (2) **DDPG(n)**: the sequential (no parallelization) version of DDPG (Lillicrap et al., 2015) implementation with double Q learning and $n$-step returns, (3) **SAC(n)**: the sequential version of SAC (Haarnoja et al., 2018) implementation with double Q learning and $n$-step returns. We run each experiment with five random seeds and plot their mean and standard deviation.

### 4.1 SETUP

**Tasks**   We experiment with six Isaac Gym benchmark tasks (Makoviychuk et al., 2021): *Ant*, *Humanoid*, *ANYmal*, *Shadow Hand*, *Allegro Hand*, and *Franka Cube Stacking* (Figure 2). For more details of the tasks, we refer the readers to (Makoviychuk et al., 2021).

**Compute**   We use NVIDIA GeForce RTX 3090 GPUs as our default GPUs for the experiments unless otherwise specified. More details are shown in Table B.3.

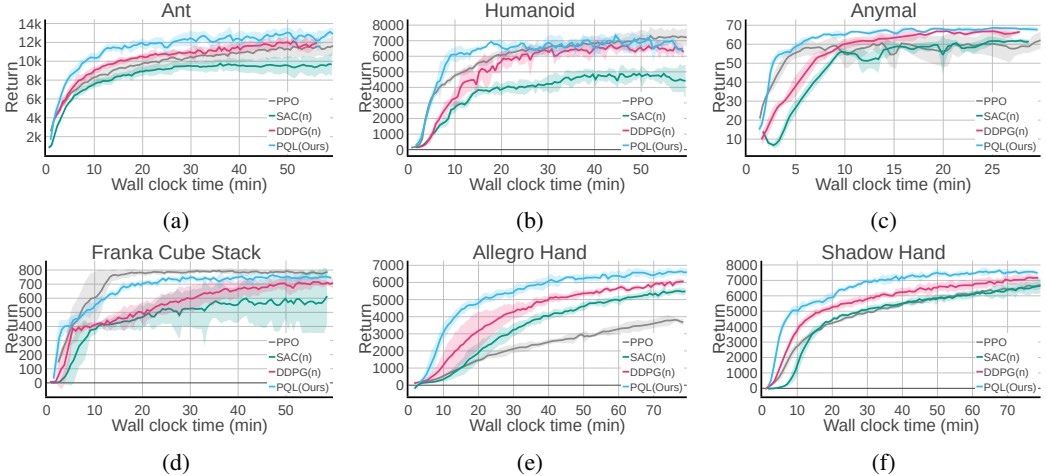

Figure 3: We compare our method to the SOTA RL algorithms (PPO, SAC with $n$-step returns, DDPG with $n$-step returns). We use $4096$ environments for training in all tasks except the PPO baseline on *Shadow Hand* and *Allegro Hand* tasks, where we use $16384$ as it gives the best performance for PPO on these two tasks as shown in Figure 4c. Our method achieves the fastest learning speed in almost all the tasks.

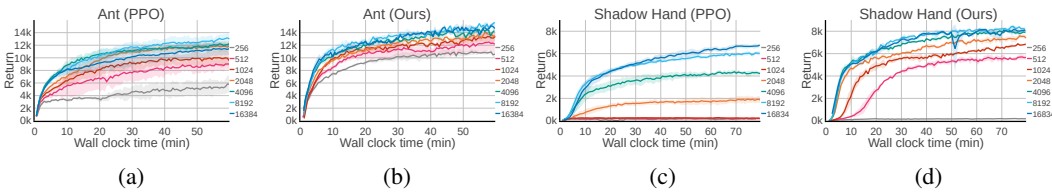

Figure 4: We sweep over different numbers of environments ($N$) on both PPO and PQL (our method). Overall, PQL is less sensitive to the number of environments than PPO on both tasks.

## 4.2 DOES PQL REDUCE THE WALL-CLOCK TIME FOR POLICY LEARNING?

As shown in Figure 3, our method (PQL) achieves the fastest policy learning in five out of six tasks compared to all the baselines. We can see that PQL learns faster than DDPG(n) on all tasks, demonstrating that our parallel scheme can speed up policy learning. In addition, we also found that DDPG(n) outperforms SAC(n) in all tasks. This might be because the exploration scheme in DDPG can be scaled up better than the one in SAC. In DDPG, we add state-independent Gaussian noise to a deterministic policy output, while the exploration solely comes from sampling in the stochastic policy distribution, which can be heavily affected by the quality of the policy distribution.

## 4.3 HOW DOES THE NUMBER OF ENVIRONMENTS $N$ AFFECT POLICY LEARNING?

GPU simulation allows for running thousands of environments in parallel on a single workstation. We anticipate that GPU simulation is only going to improve with time. However, more parallel environments will be only be useful if RL algorithms are able to exploit such data — in other words only if performance scales with more data. We therefore investigated how different algorithms scale with the number of environments. As shown in Figure 4, on the simple task (*Ant*), the learning performance is less sensitive to $N$. However, on the hard task (*Shadow Hand*), PPO's learning performance substantially drops as we decrease the number of environments. In contrast, on both tasks, our method demonstrates a stable and similar learning with all the different number of environments except when $N$ is very small ($N = 256$) on *Shadow Hand*, suggesting PQL is more robust to $N$.

## 4.4 EFFECTS OF $\beta_{p:v}$ AND $\beta_{a:v}$

As we discussed in Section 3.2, explicitly controlling the $\beta_{a:v}$ and $\beta_{p:v}$ can help reduce the variance in the learning performance under different training conditions such as fluctuated hardware utilization. If $\beta_{p:v}$ is bigger, the policy updates more frequently than the value functions, potentially

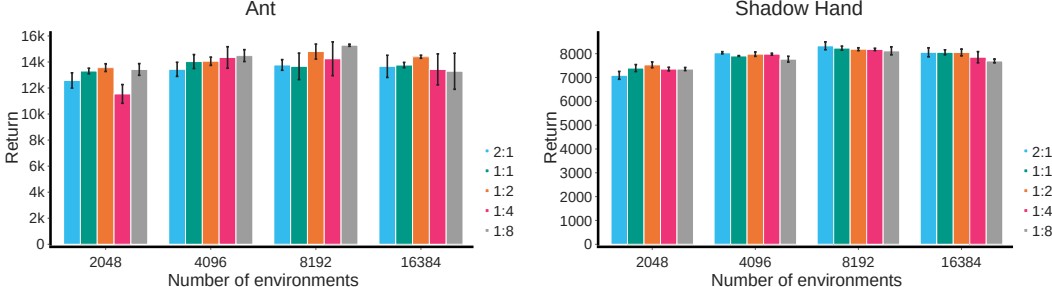

Figure 5: We show the averaged returns in evaluation after a fixed amount of training time $\Delta T$. Across the set of different number of environments we experimented (2048, 4096, 8192, 16384), we found that setting $\beta_{p:v} = 1 : 2$ generally works well. $\Delta T = 60$mins for *Ant*, and $\Delta T = 80$mins for *Shadow Hand*. The complete learning curves are in Figure C.5.

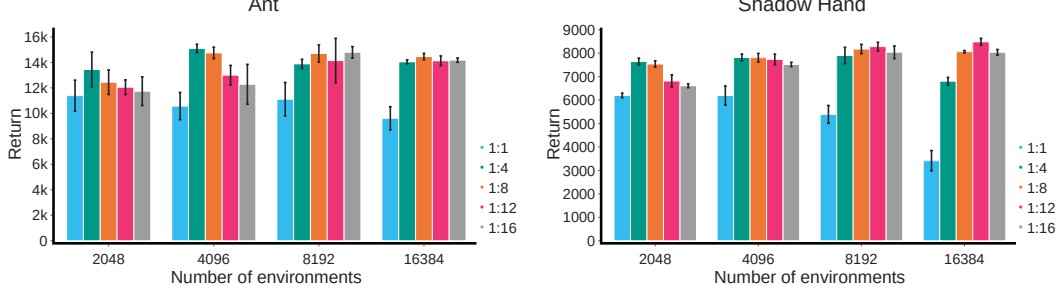

Figure 6: Given different values of $N$, we show the effect of different $\beta_{a:v}$. An overall trend we observe is that as $N$ gets bigger, it's more beneficial to update the critic more frequently. We also found that $\beta_{a:v} = 1 : 8$ generally works well given different $N$ values. So one can set $\beta_{a:v} = 1 : 8$ as a good initial value, and tune it if necessary on new tasks.

leading to policy overfitting to the stale value function, leading to bad exploration. If the policy updates much slower than the value function, then the policy might lag behind the value function a lot, which hurts the learning speed. Similarly, if $\beta_{a:v}$ is bigger, the **V-learner** might need to wait for **Actor** to collect enough data as the simulation speed cannot be changed, leading to slower learning. If $\beta_{a:v}$ is smaller, the value function updates more given the generated rollout data. To further see the effects of different $\beta_{a:v}$ and $\beta_{p:v}$, we sweep over different values on these two hyper-parameters and compare them in Figure 5 and Figure 6. Figure 5 shows that $PQL$ is relatively robust to a wide range of $\beta_{p:v}$ values. We use $\beta_{p:v} = 1 : 2$ as the default value in our experiments shown in the paper. Figure 6 shows that $\beta_{a:v}$ has a greater impact on the learning performance. An overall trend is that if we increase the number of environments, then we need to have **V-learner** update the $Q$ functions more times. For example, on *Shadow Hand*, $\beta_{a:v} = 1 : 4$ performs the best when $N = 2048$ and $N = 4096$, $\beta = 1 : 12$ performs the best when $N = 8192$ and $N = 16384$. We use $\beta_{a:v} = 1 : 8$ by default as it achieves a good performance across different $N$ values. In summary, Figure 5 and Figure 6 show that $\beta_{p:v}$ and $\beta_{a:v}$ do affect the performance with a varied number of environments. We suggest setting $\beta_{p:v} = 1 : 2, \beta_{a:v} = 1 : 8$ as a good starting point for new tasks and tune them if necessary, as these are the values we found work well on six different tasks with a different number of environments.

### 4.5 How well does mixed exploration perform?

With massively parallel simulation, one can deploy different exploration strategies in different environments to generate more diverse exploration trajectories. We use a simple mixed exploration strategy as described in Section 3.3. To see its effectiveness, we compare it to the cases where all the environments use the same exploration capacity (the same $\sigma$ values). We experimented with $\sigma \in \{0.2, 0.4, 0.6, 0.8\}$. As shown in Figure 7, the learning performance is significantly affected by the choice of $\sigma$ value. If we use the same $\sigma$ value for all the environments, then we need to tune $\sigma$ for every task. In contrast, if we use the mixed exploration strategy where every environment use a different $\sigma$ value, the agent outperforms (learn faster or at least as fast) all the other fixed $\sigma$ values. It implies that using the mixed exploration strategy can reduce the tuning effort on $\sigma$ values per task.

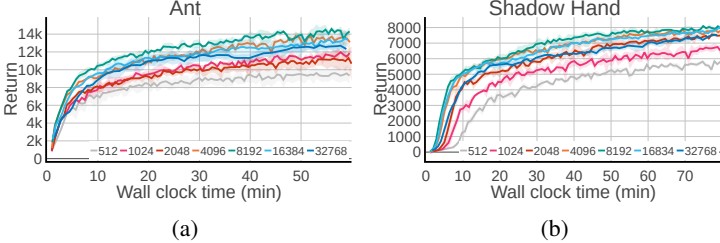

(a)              (b)              (c)              (d)

Figure 7: We compared our proposed mixed exploration scheme with applying different constant maximum noise values. We can see that the mixed exploration scheme either outperforms or is on par with other schemes, which can save the tuning effort on the noise level.

Figure 8: Effect of different batch size. Small batch size usually leads to slower learning. If the batch size is too big, the policy learning can get slowed down because GPUs have a limited amount of cores and it takes more time to process a very big batch of data.

### 4.6 Effect of Batch Size

With many parallel environments ($N$), a lot of data is quickly generated. While one can easily increase $N$ from 100s to 10,000s in Isaac Gym on a single GPU, it is infeasible to increase the replay buffer size by 100 times due to the limited GPU memory or CPU RAM (if the data is stored on CPU). Consequently, the replay buffer is overwritten frequently — meaning each collected sample may not be used efficiently. One way to efficiently utilize large amounts of changing data is to increase the batch size. To test how much increase in batch size is necessary for Q-learning with a limited capacity replay buffer to take advantage of the large amounts of incoming data, we investigated the relationship between performance and batch size. Hence, we also investigate the effects of different batch size $B$. There have been many prior works showing that using a large batch size can improve the network performance such as in contrastive learning settings (Grill et al., 2020; Chen et al., 2020). In our work, we also found that large-batch training can notably improve the learning speed in off-policy reinforcement learning for massively parallel simulation as shown in Figure 8. However, if the batch size is too big, then the learning speed can be slowed down. It is because GPUs have a limit number of CUDA cores, and it takes more time to process a very big batch data once the batch size is above some threshold value, which is another underlying trade-off.

### 4.7 Effects of Other Factors

**GPU hardware** The simulation speed and network training speed vary across different GPU models. In Table B.3, we list how much time it takes for the simulator to generate 1M environment interaction data with 4096 parallel environments on four machines with different GPU models. In our test, the simulation speed on different GPU models is as follows: GeForce 3090 > Tesla A100 > Tesla V100 > GeForce 2080Ti. We test PQL performance on all these four different machine configurations (Table B.3). In Figure 9, we can see that different GPU models affect the policy learning speed, especially on complex tasks like *Shadow Hand* which takes more simulation time.

**Number of GPUs** Our PQL framework is flexible regarding the number of GPUs available on a workstation. In other words, **Actor**, **P-learner**, and **V-learner** can be placed on any GPU on a workstation. To see how much performance variation it creates when we place the three components on different number of GPUs, we experimented with three scenarios on Tesla A100 GPUs: (1) place all three processes on the same GPU, (2) place the **Actor** on one GPU, **P-learner** and **V-learner** on another GPU, (3) place the **Actor**, **P-learner**, **V-learner** on a different GPU respectively. In the two-GPU case, the reason why we allocate **Actor** with a dedicated GPU is because the simulation for many tasks with a large number of environments can cause very high (almost full) GPU utilization.

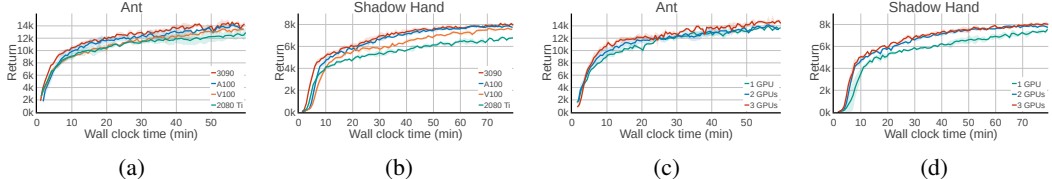

Figure 9: **(a)** and **(b)**: effect of GPU models used for running PQL. Overall, we see that PQL works robustly across different GPU models, and running on newer GPUs tend to give a faster learning. **(c)** and **(d)**: effect of number of GPUs used for running PQL. PQL can be deployed on a flexible number of GPUs. In complex tasks such as *Shadow Hand*, it is beneficial to have at least 2 GPUs where the **Actor** runs on a separate GPU as the simulation itself consumes more GPU compute as the task complexity increases.

As shown in Figure 9, our PQL framework works well in all three scenarios with one, two, or three GPUs. When the task becomes more complex like *Shadow Hand*, the simulation takes much more computation and time. Putting all three processes will slow down each one of them as the GPU utilization will be full all the time, which is why we see a bigger gap between the 2-GPU or 3-GPU training and 1-GPU training on *Shadow Hand*. In this case, it is beneficial to place the **Actor** on one GPU, and put **P-learner** and **V-learner** on other GPUs.

## 4.8 VISION-BASED TASK

Simulating vision-based tasks is much slower and more demanding on the GPU as each simulation step involves both the physics simulation and image rendering. To demonstrate our framework's generality in operating in this different but practical setting of vision-based training, we consider a simple vision-based *Ball Balancing* task. Since directly learning a vision-based policy with RL is time-consuming, we use the idea of asymmetric actor critic learning (Pinto et al., 2017) to speed up vision policy learning. To reduce the bandwidth requirement, image data is compressed using the *lz4* library for fast communication between processes. More setup details are in Appendix B.3. As shown in Figure B.1, PQL achieves better sample efficiency and higher final performance than PPO with $N = 1024$ parallel environments.

## 5 DISCUSSION

We have presented a framework for scaling up off-policy methods, especially when using GPU-based simulation. Our framework achieves state-of-the-art results on the Isaac Gym benchmark tasks, both in terms of the training wall-clock time and the final performance.

Our PQL framework provides a mechanism to balance and control the speed in different processes (data collection, policy network update, and $Q$ functions update), which leads to more stable performance across different hardware conditions or when the GPU resource is limited. While PPO requires a large number of environments to work on complex tasks such as *Shadow Hand*, PQL is more lenient on the number of environments and works well on a wide range of different numbers of environments. With a large number of parallel environments, it's beneficial to use a big batch size for training agents with a caveat that if the batch size is too big, it might take the GPU more time to process the batch data and lead to a slowdown in policy learning. We also found using different exploration scales in different environments achieves better or similar performance compared to carefully-tuned same exploration scale in all environments, which means we need less hyperparameter tuning. Even though the number of environments is $1000\times$ more compared to many $Q$-learning based agents, we did not find it necessary to use a replay buffer that is $1000\times$ bigger. In fact, a replay buffer with a capacity of 5M transitions is sufficient for our experiments even with 16843 parallel environments. Our framework's requirement for hardware is also flexible and works well with different numbers of GPUs and various GPU models.

We do not use techniques such as prioritized experience replay, dueling networks, distributional RL, or noisy nets for exploration in our work. Nonetheless, our PQL framework already gets better performance and faster learning than PPO. Whether adding these extra techniques can further improve the learning speed or final policy performance remains to be explored in future work.

## 6 REPRODUCIBILITY STATEMENT

We describe our framework in detail in Section 3.1. Our mixed exploration strategy also reduces the need for hyper-parameter turning on the exploration noise level (Section 4.5). Details of all the hyper-parameters are in Appendix B.1. We have used the same set of hyper-parameters for all the tasks. Each curve shown in the paper is obtained after running the experiment with five different random seeds. We have also investigated the performance variation on different GPU models and different numbers of GPUs in Section 4.7. Lastly, we plan to release the code in the future.

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

## APPENDIX A  PSEUDO CODE

---
**Algorithm 1 Actor** Process (main process)
---
**for** $n = 1 : W_a$ **do**
    $\pi \leftarrow$ policy network from **P-learner** process
    Initialize an empty buffer $B = \phi$
    **for** $t = 1 : H$ **do**
        $\boldsymbol{a}_t \leftarrow \pi(\boldsymbol{s}_t)$ with mixed exploration noise
        $(\boldsymbol{r}_t, \boldsymbol{s}_{t+1}) \leftarrow$ **envs**.step$(\boldsymbol{a}_t)$
        $B = B \cup \{\boldsymbol{s}_t, \boldsymbol{a}_t, \boldsymbol{r}_t, \boldsymbol{s}_{t+1}\}$
    **end for**
    $Q_1, Q_2 \leftarrow$ Q functions from **V-learner** process
    send $B, \pi$ to **V-learner**, send $\{s_t\}$ in $B$, $Q_1, Q_2$ to **P-learner**
    sleep for $t_a$ seconds to satisfy $\beta_{a:v}$
**end for**

---
**Algorithm 2 P-learner** Process
---
Initialize an empty buffer $B_p = \phi$
**for** $n = 1 : W_p$ **do**
    **if** new data received **then**
        $\{s_t\} \leftarrow$ from **Actor** process
        $Q_1, Q_2 \leftarrow$ from **Actor** process
        $B = B \cup \{s_t\}$
    **end if**
    sample a batch of $\{s_t\}$
    update $\pi$ by maximizing the $\min_{i=1,2} Q_i(s_t, \pi(s_t))$
    sleep for $t_p$ seconds to satisfy $\beta_{p:v}$
**end for**

---
**Algorithm 3 V-learner** Process
---
Initialize an empty buffer $B_v = \phi$
**for** $n = 1 : W_v$ **do**
    **if** new data received **then**
        $\{s_t, a_t, r_t, s_{t+1}\} \leftarrow$ from **Actor** process
        $\pi \leftarrow$ from **Actor** process
        $Q_1, Q_2 \leftarrow$ from **Actor** process
        $B = B \cup \{s_t\}$
    **end if**
    sample a batch of $\{s_t, a_t, r_t, s_{t+1}\}$
    update $Q_1, Q_2$ by minimizing the mean-squared Bellman error (with Double Q-learning)
    sleep for $t_v$ seconds to satisfy $\beta_{p:v}, \beta_{a:v}$
**end for**

---

## APPENDIX B  TRAINING SETUPS

### B.1  HYPER-PARAMETERS

We use the hyper-parameter values shown in Table B.1 and the reward scaling shown in Table B.2 for all the experiments unless otherwise specified. As for PPO, we use the same hyperparameter setup in Makoviychuk et al. (2021).

### B.2  HARDWARE CONFIGURATIONS

Table B.3 lists the hardware configurations of the workstations we used for the experiments. We use the machines with GeForce RTX 3090 for experiments by default. We also measure how much time

Table B.1: Hyper-parameter setup for six Isaac Gym benchmark tasks

| Hyper-parameter | PQL(ours) | DDPG | SAC |
|---|---|---|---|
| Num. Environments | 4,096 | 4,096 | 4,096 |
| Critic Learning Rate | $5 \times 10^{-4}$ | $5 \times 10^{-4}$ | $5 \times 10^{-4}$ |
| Actor Learning Rate | $5 \times 10^{-4}$ | $5 \times 10^{-4}$ | $5 \times 10^{-4}$ |
| Learnable Entropy Coefficient | - | - | True |
| Optimizer | Adam | Adam | Adam |
| Target Update Rate ($\tau$) | $5 \times 10^{-2}$ | $5 \times 10^{-2}$ | $5 \times 10^{-2}$ |
| Batch Size | 8,192 | 8,192 | 8,192 |
| Num. Epochs ($\beta_{a:v}$) | 8 | 8 | 8 |
| Discount Factor($\gamma$) | 0.99 | 0.99 | 0.99 |
| Normalized Observations | True | True | True |
| Gradient Clipping | 0.5 | 0.5 | 0.5 |
| Exploration Policy | Mix | Mix | - |
| $N$-step target | 3 | 3 | 3 |
| Warm-up Steps | 32 | 32 | 32 |
| Replay Buffer Size | $5 \times 10^6$ | $5 \times 10^6$ | $5 \times 10^6$ |

Table B.2: Reward scale

| | Reward scale |
|---|---|
| Ant | 0.01 |
| Humanoid | 0.01 |
| ANYmal | 1.0 |
| Franka Cube Stacking | 0.1 |
| Allegro Hand | 0.01 |
| Shadow Hand | 0.01 |
| Ball Balance | 0.1 |

it takes for the simulator to generate 1M interaction data with $4096$ parallel environments on *Ant* and *Shadow Hand*. We generate 1M data via the following command.

```
for i in range(244):
    action = torch.randn((4096,
                          envs.action_space.shape[0]),
                          device='cuda')
    out = envs.step(action)
```

Table B.3: Hareware configurations on different workstations

| | | Workstation 1 | Workstation 2 | Workstation 3 | Workstation 4 |
|---|---|---|---|---|---|
| CPU | | AMD Threadripper 3990X | Intel Xeon Gold 6248 | AMD Rome 7742 | Intel Xeon W-2195 |
| GPU | | GeForce RTX 3090 | Tesla V100 | Tesla A100 | GeForce RTX 2080 Ti |
| GPU CUDA Cores | | 10496 | 5120 | 6912 | 4352 |
| GPU FP32 TFLOPs | | 35.58 | 16.4 | 19.5 | 13.45 |
| Time for generating | Ant | $1.678 \pm 0.006$ | $2.117 \pm 0.038$ | $1.999 \pm 0.004$ | $3.397 \pm 0.014$ |
| 1M data ($N = 4096$) (s) | Shadow Hand | $6.706 \pm 0.028$ | $9.051 \pm 0.035$ | $8.653 \pm 0.101$ | $10.885 \pm 0.025$ |

## B.3 VISION EXPERIMENT SETUP

We render the RGB camera image in a resolution of $48 \times 48$. The CNN part of our vision network $g(o_t)$ is as follows:

```
Conv(3,32,3,2)-BN(32)-ReLU-3x(Conv(32,32,3,2)-BN(32)-ReLU)
```

where `Conv(a,b,k,s)` is a Convolutional layer with input channels $a$, output channels $b$, kernel size $k$, stride $s$.

Since our policy input contains a history of observations $(o_{t-2}, o_{t-1}, o_t)$, we use the same CNN to extract the feature of each observation and then concatenate all the embeddings. Then, the concatenated embedding goes through an MLP network $h$:

```
FC(256)-ReLU-FC(63)-ReLU-FC(3)
```

In summary, at each time step $t$, the policy output is $h[\text{cat}(g(o_{t-2}), g(o_{t-1}), g(o_t))]$. Storing images in a replay buffer can take up a lot of memory. Therefore, we experiment with different placements of the replay buffer: (1) put the replay buffer on a GPU with a big memory, (2) put the replay buffer on CPU RAM. We use the same A100 GPUs for all these image-based experiments. Figure B.1 shows that our method (PQL) works with either the replay buffer on the GPU or CPU, and it achieves much faster learning and better performance than PPO.

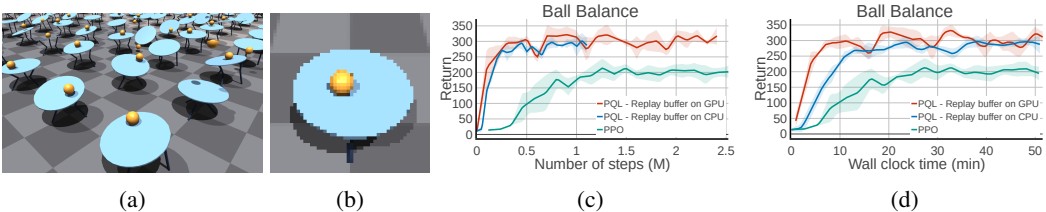

|   (a)   |   (b)   |   (c)   |   (d)   |

Figure B.1: **(a)**: the *Ball Balancing* task in Isaac Gym. **(b)**: the rendered RGB image from the simulated camera. **(c)** shows the learning curves regarding the number of environment steps. **(d)** shows the training wall-clock time. We can see that PQL achieves both better sample efficiency and higher final performance than PPO.

Table B.4: Hyper-parameter setup for the *Ball Balancing* task.

| Hyper-parameter | PQL(ours) | PPO |
| --- | --- | --- |
| Num. Environments | 1,024 | 1,024 |
| Critic Learning Rate | $5 \times 10^{-4}$ | $5 \times 10^{-4}$ |
| Actor Learning Rate | $5 \times 10^{-4}$ | $5 \times 10^{-4}$ |
| Optimizer | Adam | Adam |
| Target Update Rate ($\tau$) | $5 \times 10^{-2}$ | - |
| Batch Size | 4,096 | 4,096 |
| Horizon length | 1 | 16 |
| Num. Epochs | 12 | 5 |
| Discount Factor($\gamma$) | 0.99 | 0.99 |
| Normalized Observations | True | True |
| Gradient Clipping | True | True |
| Exploration Policy | Mix | - |
| $N$-step target | 3 | - |
| Warm-up Steps | 32 | - |
| Replay Buffer Size | $10^6$ | - |
| Clip Ratio | - | 0.2 |
| GAE | - | True |
| $\lambda$ | - | 0.95 |

## APPENDIX C   ADDITIONAL EXPERIMENTS

**$n$-step returns**   We investigate how much does $n$-step returns help for PQL. As shown in Figure C.2, adding $n$-step return leads to faster learning than not using $n$-step return ($n = 1$). However, using a big $n$ value hurt the learning. Empirically we found that $n = 3$ gives us the best performance.

**Replay buffer size** We investigate how the replay buffer size $|B|$ affect the policy learning performance. We experimented with replay buffers with a size of $\{1, 5, 10, 20\}$M. The replay buffers stay on GPUs. As shown in Figure C.2, in all cases, policies can learn well. In addition, $|B| \in \{1, 5\}$ lead to faster policy learning in the beginning of the training than $|B| = \{10, 20\}$. We hypothesize this is because with a smaller replay buffer, the old and less informative samples get replaced much faster which is more important in the early stage of the training. On the other hand, as shown in Table B.3, generating 1M interaction data with Isaac Gym takes a very small amount of time. For example, on a GeForce 3090 GPU, it only takes 1.7s to generate 1M data on *Ant*, and 6.7s on *Shadow Hand*. This implies that with massively parallel simulation, the data in the replay buffer becomes more on-policy than in most $Q$-learning agents. We hypothesize that PQL, DDPG still works well in this case because the large amount of parallel environments can generate diverse enough data in a short period of time.

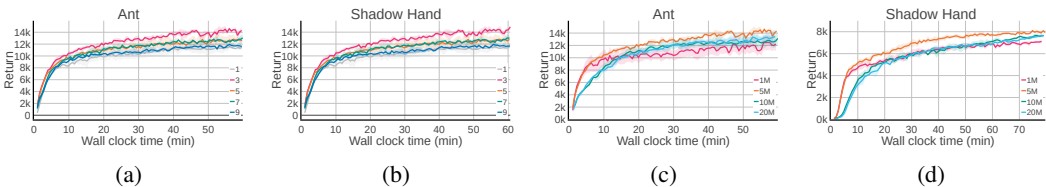

Figure C.2: **(a)** and **(b)**: effect of $n$-step return. $n = 3$ performs the best. **(c)** and **(d)**: effect of different replay buffer size.

**PQL for SAC** As discussed above, PQL framework is flexible and can be combined with different $Q$-learning methods. Here, we show that PQL can be combined with SAC as well. Figure C.3 shows that adding the PQL framework to SAC substantially speeds up the learning speed of SAC.

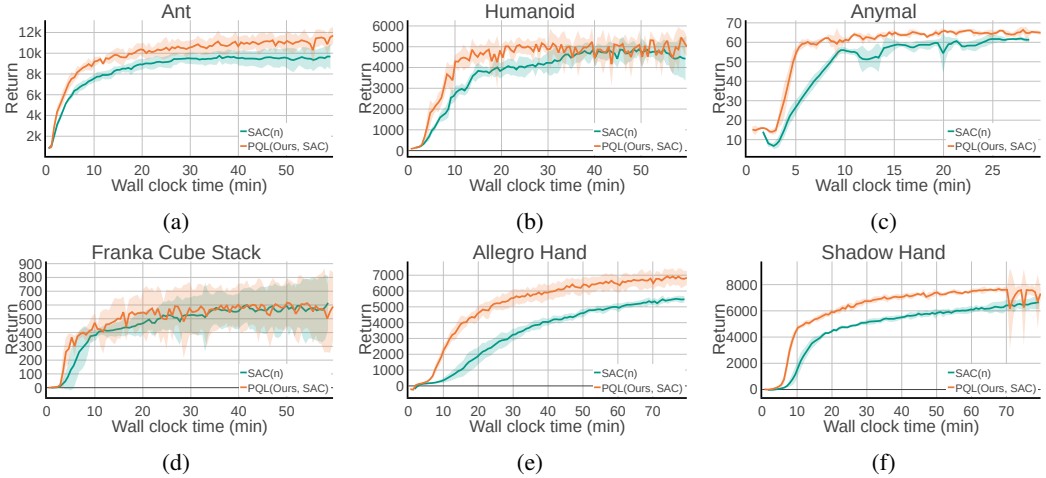

Figure C.3: We apply our parallel $Q$-learning to SAC. PQL + SAC achieves faster learning than SAC itself.

**Sample efficiency compared to baselines** Figure C.4 shows the sample efficiency of each algorithm on different environments. Overall, we see that PQL achieves the best sample efficiency. In addition, DDPG(n) also outperforms SAC(n) in terms of the sample efficiency on these tasks.

**Sweep over different $\beta_{a:v}$ and $\beta_{p:v}$** Figure C.5 shows the complete learning curves with different $\beta_{p:v}$ values and different number of environments. Similarly, Figure C.6 shows the learning curves for different $\beta_{a:v}$.

**Comparison of our implementation with RL-games** In this work, we implemented all the algorithms (PQL and all the baselines) from scratch, as it gives us the most flexibility in exploring

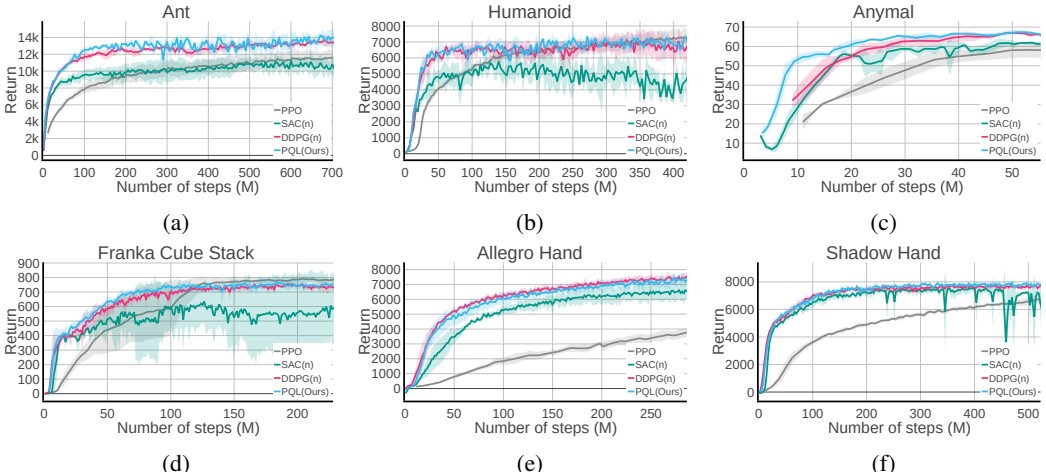

Figure C.4: Similar to Figure 3, we show that our method (PQL) also achieves better sample efficiency than baselines.

different design choices that can affect learning performance. To show that our codebase provides good performance, we compare it against the most commonly used RL codebase used for Isaac Gym, which is RL-games (Makoviichuk & Makoviychuk, 2022). However, RL-games only support PPO and SAC. Hence, we compare our implementations of PPO and SAC against the ones in RL-games.

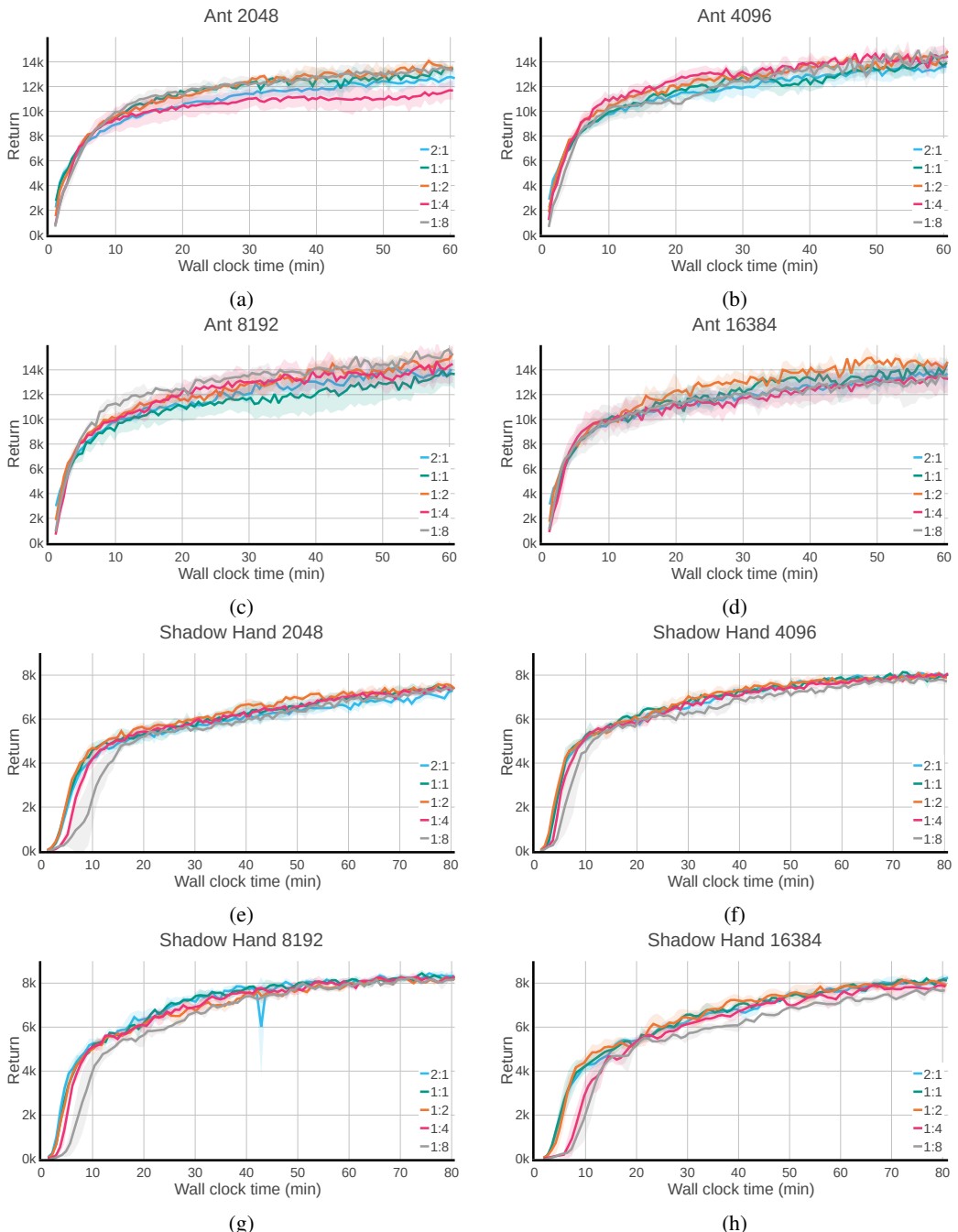

Figure C.5: Learning curves for different $\beta_{p:v}$.

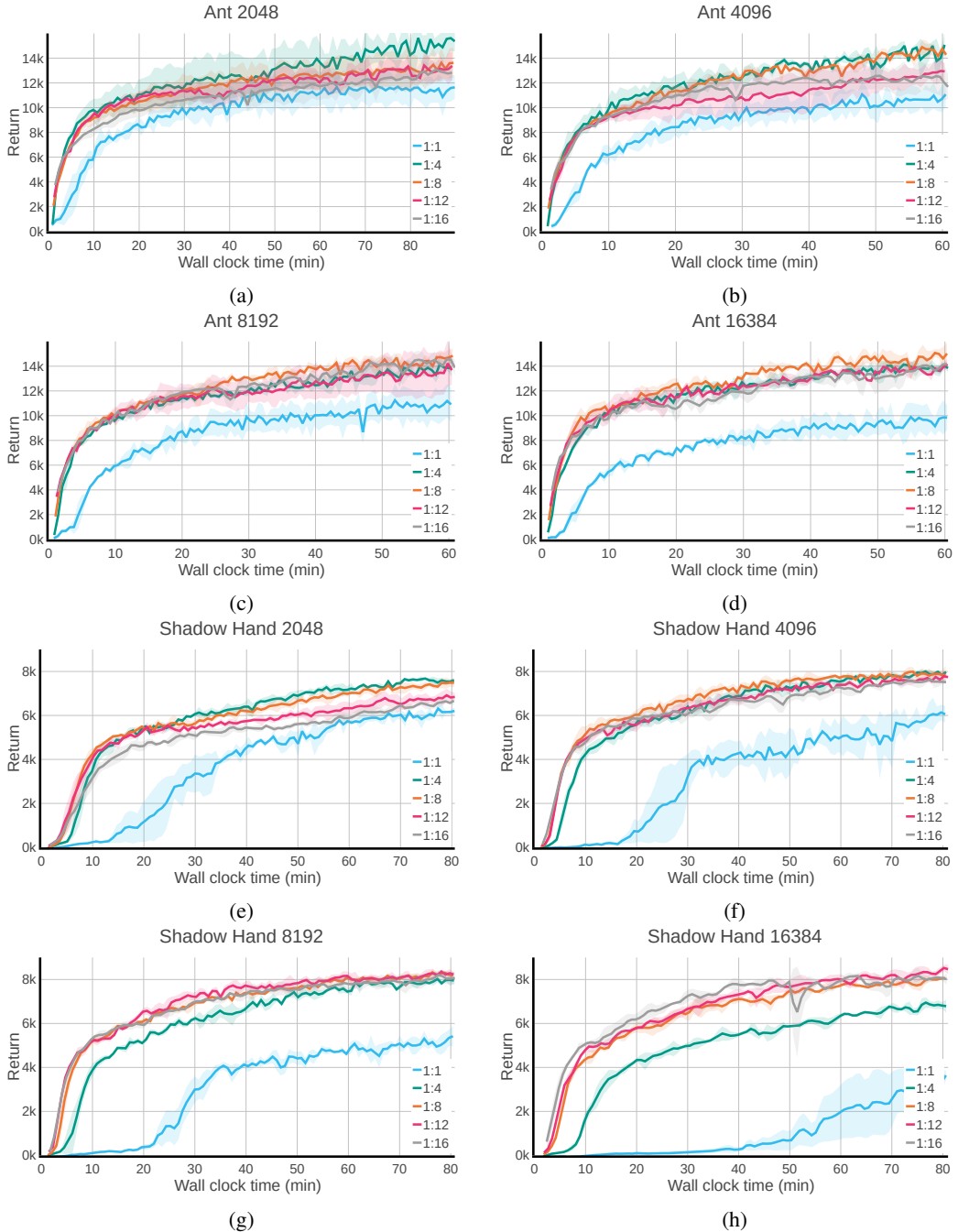

Figure C.6: Learning curves for different $\beta_{a:v}$.

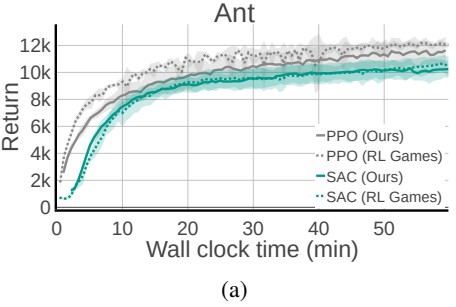
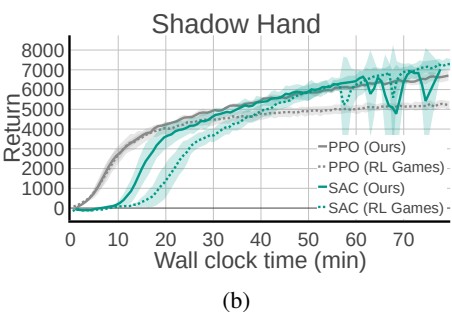

(a)             (b)

Figure C.7: Comparison between our implementations of PPO and SAC against the ones provided in RL-games. We can see that both codebases provide similar performance, showing that our implementation is good and reliable. On *Shadow Hand*, our PPO learns even faster and better than the PPO in RL-games.

