# OpenReview forum: "Parallel $Q$-Learning: Scaling Off-policy Reinforcement Learning"
_ICLR.cc/2023/Conference — Submitted to ICLR 2023_

### Official Review · Reviewer_iU6x · 2022-10-23

**Confidence:** 4
**Correctness:** 2
**Technical Novelty And Significance:** 2
**Empirical Novelty And Significance:** 2
**Recommendation:** 5

**Clarity, Quality, Novelty And Reproducibility:**

The writing of this paper has a considerable space to be improved. The novelty of the proposed approach over existing methods requires further elaborations. The reproducibility is good.

**Details Of Ethics Concerns:**

There are no ethics concerns.

**Strength And Weaknesses:**

Strength:
1. The evaluation of the proposed method is extensive and involves many different tasks.
2. The parallel simulation environments considered here are interesting and they might be important to greatly reduce the training costs for RL.

Weaknesses:
1. The introduction of the three parallel processes is quite abrupt without adequate discussions about the motivation of such considerations. It's unclear why these design choices are inspired from existing methods like DDPG or other variants. Furthermore, what's the challenges to design a Q-learning algorithm for parallel environments? Why a direct extension of DDPG can not work?
2. It's not clear what's the key difference between parallel environments on a single workstation and parallel environments on a computer cluster. Does such difference really matters in the design of the algorithm? Why?
3. The discussion of the main techniques is too brief and can not convince readers the design rationales behind the proposed algorithm.
4. Another suggestion is to write the proposed method as an algorithm so that it would be more clear the algorithmic improvements made over existing approaches. Stressing the design considerations that make the proposed approach more efficient than others.

**Summary Of The Paper:**

Motivated by the data inefficiency problem of off-policy algorithms, like PPO, this paper proposed a parallel version of the Q-learning algorithm. Specifically, this approach uses three parallel processes, namely actor, V-learner and P-learner, to perform a DDPG-like learning process. Empirical evaluations on six Isaac Gym tasks has demonstrated that the proposed approach has better learning efficiency than baselines like PPO, SAC, etc.

**Summary Of The Review:**

Despite an extensive amount experiments have been conducted, the writing of this paper requires a considerable revisions to convince the contributions of this paper and illustrate how each design consideration leads to performance improvement over existing methods.

---

> ### Author Response · Authors · 2022-11-19
> **Response to Reviewer iU6x (1/2)**
>
> Thank you for your insightful feedback. We would like to address your concerns as follows.
>
> > The introduction of the three parallel processes is quite abrupt without adequate discussions about the motivation of such considerations. It's unclear why these design choices are inspired from existing methods like DDPG or other variants.
>
> We have included a paragraph at the beginning of the Method section to better motivate the design choices in the PQL framework. In short, off-policy methods like DDPG have a few advantages over on-policy methods such as PPO.
>
> * Off-policy methods can be more data-efficient than on-policy methods, implying a possibility of speedup over on-policy methods in training if we can reduce the wall-clock time for off-policy methods. Such a speedup gained from sample efficiency is expected to be more significant as the task complexity increases (for example, tasks that involve more contacts between bodies, leading to slower simulation). For example, as shown in Figure 3, the advantage of using PQL over PPO is much more salient on challenging contact-rich tasks such as Allegro Hand and Shadow Hand.
> * Second, with an off-policy algorithm, it is much easier to parallelize the data collection, policy update, and value update as there is no requirement for the training data to be on-policy as in on-policy methods. Paralleling the data collection and network update allows each one of them to run asynchronously, greatly improving the throughput of the data generation and consumption and leading to faster training.
>
> > Furthermore, what's the challenges to design a Q-learning algorithm for parallel environments? Why a direct extension of DDPG can not work?
>
> * First, our PQL framework innovates on parallelizing all three components in DDPG (data collection, value update, and policy update). Such a parallelization greatly improves the training wall-clock time upon the sequential version of DDPG. Another technique we use to remove the hyperparameter tuning on the amount of exploration noise is *mixed exploration*. We use different exploration noises in different environments. As shown in Fig 7, with mixed exploration, we are able to achieve good performance on all tasks without tuning the noise values.
> * Second, we also show in the paper (Fig. 3) that with proper engineering (n-step return, double Q learning, etc.), DDPG also works for the Isaac Gym tasks. But PQL further speeds up DDPG. Till now, no open-source library has demonstrated a good implementation of DDPG that works for Isaac Gym tasks. We also found that DDPG works much better than SAC in this case. We implemented all the algorithms used in the paper from scratch and verified that our implementation provided good performance. As shown in Fig. C.6, our implementation of SAC and PPO gets better performance than the most widely used RL library, rl-games, for solving Isaac Gym tasks.

---

> > ### Author Response · Authors · 2022-11-19
> > **Response to Reviewer iU6x (2/2)**
> >
> > > It's not clear what's the key difference between parallel environments on a single workstation and parallel environments on a computer cluster. Does such difference really matters in the design of the algorithm? Why?
> >
> > First of all, from a practical perspective, in terms of the simulation environments themselves, it makes a huge difference between GPU-based simulators such as Isaac Gym and CPU-based simulators such as Mujoco. The former can parallelize thousands of environments on a single GPU, and the latter would require a compute cluster.  Developing a good framework in the era of GPU-based simulation is of great practical value that can speed up research in challenging robotics tasks.
> >
> > Second, the number of parallel environments that can be provided by Isaac Gym on a single GPU is much greater than 1000 (we have experimented with 16,384 parallel environments), which is orders of magnitude bigger than the number of parallel environments used in many prior distributed off-policy framework such as APEX (the maximum we found in APEX paper is 256 environments). With many more parallel environments, many new questions arise. For example, **how can we make the best use of the tons of interaction data generated at each step? How do we balance the data generation speed and network update speed? How does** $N>>1000$ ($N$, the number of environments) affect the learning performance? **Do we need a much bigger replay buffer?** These are the questions we try to address in the paper. We did extensive experiments to show how to adjust the ratio between the data collection speed and policy/value function update speed (Fig 5 and Fig 6). This leads to insights such as with more environments, it is usually beneficial to update the critic function more frequently. We also show that even with tens of thousands of parallel environments, a replay buffer of 5M capacity is still enough. We also investigated the effect of batch size, different exploration schemes, etc. We believe all these new insights (which have not been shown in prior works in this setting) can provide good guidance on designing off-policy algorithms and how to set some of the critical hyperparameters given $>10K$ parallel environments.
> >
> > > The discussion of the main techniques is too brief and can not convince readers the design rationales behind the proposed algorithm.
> >
> > We would like to kindly refer the reviewer to our answer to your first comment.
> >
> > > Another suggestion is to write the proposed method as an algorithm so that it would be more clear the algorithmic improvements made over existing approaches. Stressing the design considerations that make the proposed approach more efficient than others.
> >
> > Thanks for the suggestions. We have included the pseudo-code in Appendix A. Also, we have included a motivation/intuition paragraph at the beginning of the Method section.

---

### Official Review · Reviewer_BMNj · 2022-10-23

**Confidence:** 3
**Correctness:** 4
**Technical Novelty And Significance:** 3
**Empirical Novelty And Significance:** 3
**Recommendation:** 8

**Clarity, Quality, Novelty And Reproducibility:**

The paper is written clearly, and the research question is interesting with high relevance. The novelty appears to be non-trivial, and the results are detailed enough for reproduction.

**Strength And Weaknesses:**

Strength:

1. The paper is written very clearly, with all the sections accurately describing what the algorithm is about. The relationship between the different components in the framework has been clearly demonstrated by graphs, and the intuition behind the components as well as choices of returns/RL structures is easy to follow.

2. The experiments are comprehensive, and provide sufficient details for reproducibility as well as ablation studies. In particular, the discussion on the ratio of beta values(indicating update frequency comparisons) is very informative.

Weaknesses:

1. The algorithm is named Parallel Q-learning. However, the key of the algorithm lies in decomposing DDPGs/SACs and making use of techniques of improving Q-learning to better estimate Q-functions in the learning process. It would be helpful to emphasize in the abstract and intro where the novelty lies since the title may be a bit misleading.

2. While the results of the experiments are largely self-contained and informative, it would be helpful to include the performances of PQL with SAC as the underlying algorithm. Since it is indicated in the paper that the PQL framework can be naturally extended to SACs, the inclusion of such ablation studies would answer the question how much the effectiveness of the PQL framework would depend on the choice of off-policy algorithms.



**Summary Of The Paper:**

This paper proposes paralleled Q-learning, a three-component framework to accelerate Q-learning. The design of the components enables concurrent execution, and experiments on Isaac Gym and vision-based benchmarks against other state-of-the-art algorithms such as PPO, DDPG and SAC show significant improvement as measured by cumulative return.

**Summary Of The Review:**

I would rate this paper a 8, given its scope of contributions and good quality.

---

> ### Author Response · Authors · 2022-11-19
> **Response to Reviewer BMNj**
>
> We would like to thank reviewer BMNj for the helpful comments. We address the reviewer's concerns as follows:
>
> > The algorithm is named Parallel Q-learning. However, the key of the algorithm lies in decomposing DDPGs/SACs and making use of techniques of improving Q-learning to better estimate Q-functions in the learning process. It would be helpful to emphasize in the abstract and intro where the novelty lies since the title may be a bit misleading.
>
> Thanks for your comments! We have updated the abstract and introduction accordingly in the revised version (shown in blue text).
>
> > While the results of the experiments are largely self-contained and informative, it would be helpful to include the performances of PQL with SAC as the underlying algorithm.
>
> We have added the implementation of PQL framework upon SAC. The results are shown in the newly added figure (Figure C.2 in the appendix). PQL + SAC achieves substantially faster learning than SAC.

---

### Official Review · Reviewer_bLp3 · 2022-10-25

**Confidence:** 3
**Correctness:** 2
**Technical Novelty And Significance:** 2
**Empirical Novelty And Significance:** 2
**Recommendation:** 3

**Clarity, Quality, Novelty And Reproducibility:**

The empirical evaluation of PQL shows that it's able to learn faster than other approaches in 6 simulated environments. However, the presentation of the paper doesn't help understand how this result is achieved. Figure B.2 indicates that PQL is roughly as sample efficient as DDPG which is expected since they use similar off-policy approaches, but more sample efficient than PPO (on-policy), which is also expected. Figure 3 shows that PQL is faster than PPO and DDPG on the same hardware, so the implementation of PQL has to be able to do more work per second. How this is achieved is not explained. Is the GPU utilization increased and if so, how ? Does the framework better balance the workload between the CPU and the GPU ? Is one of the code bases that PQL leverages more efficient than the ones leveraged in the PPO and DDPG implementations used as baselines, thus reducing the training time ?

It's not clear what figure 5&6 demonstrate. If the relative ratios of total rollouts, updates to the Q function, and policy updates had been kept constant while the number of environments changes, and if the noise levels had been modified to avoid increasing or decreasing how much exploration is performed, we could have discovered the presence of absence of a communication bottleneck in the implementation. As done in the paper, there are too many unaccounted factors to be able to draw any conclusions.

The batch size experiment needs to be better explained. How is the batch size applied ? Is it applied solely on the Q and policy networks, or is it also applied to the actors ? It would have been better to show how the efficiency of the 3 components of the approach changes as batch size varies.

Some aspects of the paper, such as the beta factors in section 3.2, are described in more detail than necessary: it's enough to define them, any reader can infer what they do from there. On the other hand, some mechanisms need more explanations. For example, why is f_a defined as the number of rollouts per environment per unit of time instead of the total number of rollouts per unit of time ? The later could make the setting more robust to changes in the number of environments operating in parallel, and could have decreases the number of hyper-parameters. Similarly, in section 3.2, how are the values of a_u and a_l determined and how would the noise injection process change if the actions were discrete instead of continuous ?

Please increase the font on the figures. The legends are very tiny, and I suspect impossible to read on a printed version of the paper.




**Strength And Weaknesses:**

Scale is a major force behind the success of reinforcement learning. However, unlike numerous previous approaches that achieve scale by distributing the computation over large amounts of servers, PQL proposes to leverage a single workstation by multiplexing the simulation of thousands of environments along with a value function and a policy on a single GPU.

The approach appears to be similar to that of DDPG, and the implementation is based on an existing framework targeted for distributed training that was repurposed to run multiple simulations of the Isaac Gym environment on a single workstation, which is the main contribution of the paper. However, several points are unclear:
 * Where do the wall-time gains come from ? Is Isaac Gym more efficient when running multiple simulations in parallel, or is there something else driving the performance improvement ? Ideally, I'd like to be able to answer the question: what aspect(s) of PQL approach is inherently better and drives most of the performance improvements.
 * How applicable is this beyond Isaac Gym ? Would it help with other simulation based gyms ?
 * How would the PQL approach compare against other distributed RL implementations such as IMPALA/seed-RL if these implementations were also repurposed to target the Isaac Gym ?

**Summary Of The Paper:**

This paper proposes to run thousands of simulations of a RL environment to speedup Q learning. Unlike previous efforts, that leverage distributed computing to run agents on large numbers of machine, they propose to leverage a single workstation running numerous tasks in parallel.
The paper contributes a novel framework, called PQL, that can train RL policies faster than previous efforts without requiring a large amount of compute resources.

**Summary Of The Review:**

PQL is an interesting paper that demonstrates that a better implementation of the Q learning strategy can make it competitive against PPO from the point of view of training time. However, the author need to provide an intuition as to why their approach is inherently more performant. Furthermore, the evaluation needs more work to clearly demonstrate how the authors achieve superior performance.

---

> ### Author Response · Authors · 2022-11-09
> **Request for clarification on the comment**
>
> We would like to thank the reviewer for the comments and suggestions. Before we answer all the concerns, we wanna make sure we understand your comment correctly. In particular, we are a bit confused about the meaning of this sentence: "If the relative ratios of total rollouts, updates to the Q function, and policy updates had been kept constant while the number of environments changes, and if the noise levels had been modified to avoid increasing or decreasing how much exploration is performed, we could have discovered the presence of absence of a communication bottleneck in the implementation. " What does the presence of absence of communication bottleneck refer to here? Do you mind elaborating on this point a bit more so that we can better understand the comment? Thanks.

---

> ### Author Response · Authors · 2022-11-19
> **Response to Reviewer bLp3 (1/5)**
>
> We sincerely thank reviewer bLp3 for the helpful feedback and suggestions on our paper. We address the reviewer's concerns below.
>
> > Where do the wall-time gains come from ? Is Isaac Gym more efficient when running multiple simulations in parallel, or is there something else driving the performance improvement ?
>
> Yes, Isaac Gym can run many more simulations in parallel compared to other simulators such as PyBullet and Mujoco. However, the performance gains in wall-clock time are not just because of using Isaac — we compare the proposed PQL algorithm against PPO, and DDPG — and it's much faster. If it were only due to the simulator, PQL wouldn’t be faster than other alternatives. The wall-clock time gains are attributed to the following:
>
> * Compared to PPO, PQL is built upon Q-Learning (particularly a modified version of DDPG), which is more data efficient than on-policy methods such as PPO. While variants of off-policy learning such as TD3 and SAC work much better than DDPG in the case of a few parallel environments, we found with massive parallelism, DDPG with double Q-learning performed as well or better. Therefore we built upon DDPG as the base Q-learning algorithm.
> * Off-policy learning allows us to decouple “data collection” and the update of value and policy networks which is essential for speed up.
> * Because the data is being generated at such high-speed, it raises many questions about Q-learning: the replay buffer, which is believed to be critical for stability, is overwritten very quickly, which may actually make Q-learning be unstable. Depending on the particular GPU one is using — the relative time between data collection/value/policy network training can vary — leading to GPU-dependent performance gains. We conducted a thorough empirical investigation guided by theoretical intuition to determine hyper-parameters and parallelization implementation that overcome these challenges. The result is PQL, which is superior to both prior Q-learning and on-policy implementations — it improves performance across environments and across choices of GPU architectures. PQL is, therefore, a plug-and-play alternative to existing RL algorithm choices.
>
> As tasks become more complex, the simulation speed decreases. In such scenarios, the gap between PQL’s and PPO’s performance increases even more — a favorable scaling for PQL. For example, Fig. 3 shows that the gap between PQL and PPO is larger such as Allegro Hand and Shadow Hand compared to simpler tasks such as Ant and Anymal locomotion.
>
> > How applicable is this beyond Isaac Gym ? Would it help with other simulation based gyms ?
>
> Our algorithm is optimized for GPU-based simulators supporting a large number of parallel environments and doesn’t make Isaac Gym specific assumptions. In recent years, GPU-based simulation has led to many breakthroughs in robotics (see Related Work for a discussion) and we believe that the trend of GPU-based simulation is only going to grow. PQL has the potential of helping researchers solve more complex problems faster. Our experiments indicate the gap between baselines and PQL is larger for more complex tasks (see Allegro and Shadow Hand tasks in Fig. 3.)
>
> > How would the PQL approach compare against other distributed RL implementations such as IMPALA/seed-RL if these implementations were also repurposed to target the Isaac Gym ?
>
> The closest distributed RL framework to ours is ApeX. However, ApeX used a maximum of 256 environments. When the number of parallel environments is in tens of thousands — different considerations/challenges come into play. Our paper investigates these differences and proposes, PQL, that is suitable for a large number of environments while still maintaining advantageous or at-par performance when the number of environments is fewer (e.g., See Figure 4(a) / (b) comparing PPO v/s PQL as we change the number of environments).
>
> As reviewers requested, we tried IMPALA on the tasks used in the paper. We experimented with the following open-source implementations:
> * torchbeast: [https://github.com/facebookresearch/torchbeast](https://github.com/facebookresearch/torchbeast)
> * Dl-engine: [https://github.com/opendilab/DI-engine](https://github.com/opendilab/DI-engine)
>
> The original IMPALA implementation is only for discrete actions and not for continuous action spaces. Despite our best attempts, we weren’t able to successfully adapt IMPALA to tasks with continuous actions. There are open GitHub issues describing the same challenge found by other users (https://github.com/ray-project/ray/issues/11733#issuecomment-721710047). The maintainers for RLlib (https://docs.ray.io/en/latest/rllib/index.html), a well-implemented RL library for distributed training, also do not recommend using IMPALA for tasks with continuous action space. E.g., one of the maintainers of RLLib explicitly comments — don’t use IMPALA for continuous action space. However, we are still trying our best, and if we succeed, we will post an update.

---

> > ### Author Response · Authors · 2022-11-19
> > **Response to Reviewer bLp3 (2/5)**
> >
> > > PQL is faster than PPO and DDPG on the same hardware, so the implementation of PQL has to be able to do more work per second. How this is achieved is not explained.
> >
> > PQL is built upon DDPG. The key is to parallelize the data collection (Actor), policy learning (P-learner), and value function learning (V-learner). In DDPG, these three components happen in sequence: data collection → value function update → policy learning → data collection … Therefore, our PQL (a parallel version of DDPG) achieves faster learning than DDPG.
> >
> > Similarly, in PPO, the data collection and policy/value function are updated in sequence to make sure the interaction data collected from the environments are on-policy — and therefore cannot be parallelized in the same way.  Furthermore, off-policy algorithms are known to be more data efficient than PPO. So PQL also outperforms PPO even in terms of training time.
> >
> > > Is the GPU utilization increased and if so, how ? Does the framework better balance the workload between the CPU and the GPU ?
> >
> > GPU utilization is increased because the data collection, policy update, and value update all run in parallel instead of sequentially. For state-based policy (the input is a low-dimensional state, instead of images) learning, we do not use CPU. All the data, including the interaction data generated by the environments and the network update, are on GPUs. Even the environments themselves (Isaac Gym) use GPUs for simulation, not CPUs.
> >
> > > Is one of the code bases that PQL leverages more efficient than the ones leveraged in the PPO and DDPG implementations used as baselines, thus reducing the training time ?
> >
> > We implemented PQL, PPO, SAC, and DDPG all from scratch because it provided us the flexibility to easily explore different design choices and adapt them to parallel simulation workflows. We compare the performance of our PPO / SAC against the most commonly used RL codebase used for Isaac Gym, which is RL-games (https://github.com/Denys88/rl_games). As shown in Fig. C.6, our and RL-games lead to similar performance, showing that our implementation of PPO / SAC is equally efficient by default and also matches the performance reported in the literature. Unfortunately, RL-games doesn’t implement DDPG, because the common belief is that SAC is better than DDPG, so we couldn’t compare. However, our experiments indicate the DDPG is a better choice when it comes to parallelizing Q-learning (see Figure 3).

---

> > > ### Author Response · Authors · 2022-11-19
> > > **Response to Reviewer bLp3 (3/5)**
> > >
> > > > It's not clear what figure 5&6 demonstrate. If the relative ratios of total rollouts, updates to the Q function, and policy updates had been kept constant while the number of environments changes, and if the noise levels had been modified to avoid increasing or decreasing how much exploration is performed, we could have discovered the presence of absence of a communication bottleneck in the implementation. As done in the paper, there are too many unaccounted factors to be able to draw any conclusions.
> > >
> > > Figure 5 and 6 show the effect of $\beta_{a:v}$ (ratio of number of environment interactions v/s number of value network updates) and $\beta_{p:v}$ (ratio of number of policy updates v/s number of value network updates)  on the learning performance. The results show that different values of $\beta_{a:v}, \beta_{p:v}$ are apt depending on how many parallel environments are used. Based on the analysis, we suggest setting $\beta_{a:v} = 1:8$ and $\beta_{p:v} = 1:2$ as a good rule of thumb.
> > >
> > > The reviewer questions if these results would be valid if instead of varying the number of environments, the amount of exploration was varied. However, we used the mixed exploration strategy — where given N levels we uniformly divide them to uniformly span gaussian noise with standard deviation in the range [0.05, 0.8]. We found this strategy to remove the need for tuning the exploration levels. Therefore, we used the same exploration strategy in Figures 5 and 6 too.
> > >
> > > We are not sure if the reviewer is also asking whether one can just increase the exploration noise to get the same effect of adding more number of environments, or whether one should reduce the noise level when $N$ gets bigger to study the effect of $\beta_{a:v}, \beta_{p:v}$. So we answer both cases here.
> > >
> > > For the first case,  adding more noise is not the same as having more parallel simulators — e.g., adding large amount of noise means that the agent essentially takes random actions, which slow down progress towards a directed path (its more akin to breadthwise search). With more parallel simulators and lesser noise per simulation, the agent can perform more directed exploration in vicinity of its policy (more akin to depth-wise search).
> > >
> > > For the second case, it is true that if we increase $N$, then the rollout data will be more diverse as there are more environments and each environment can have different exploration data. The reviewer might be suggesting that we should reduce the noise level in this case to counteract the increase in the exploration brought by the bigger $N$ so that we can have the “same” amount of exploration regardless of the number of environments. And the reviewer might think this would be a more fair setup for studying the effect of $\beta_{a:v}, \beta_{p:v}$. However, there is no way to guarantee two setups with different number of environments will have the same amount of exploration no matter how we tune the noise level. Plus, our goal is to find a good set of hyperparameters that maximize the PQL performance. If we intentionally reduce the noise level when we increase the $N$, then the benefit of having a large number of parallel environments is compromised. Moreover, we are studying the effect of $\beta_{a:v}, \beta_{p:v}$ with the mixed-exploration strategy which does not require tuning the noise level. The purpose of Figure 5 and 6 is to guide users to set $\beta_{a:v}, \beta_{p:v}$ when they apply PQL to perhaps new tasks.
> > >
> > > We whole-heartedly agree with the reviewers experiments should not have unaccounted factors. We have attempted the best to do so. Therefore we fixed the exploration strategy to the one we found to work best, and experimented with other hyperparameters (e.g., batch size). However, it is also true that we have not simultaneously performed all ablations of all hyperparameters which — an infeasible experiment for us to run. Instead, we have relied on a more computationally feasible approach of building intuition for a hyperparameter and then fixing it. E.g., we experimented with different exploration strategies until we found a satisfactory one. Afterward, which we kept this exploration strategy fixed and worked on other more critical hyper-parameters.

---

> > > > ### Author Response · Authors · 2022-11-19
> > > > **Response to Reviewer bLp3 (4/5)**
> > > >
> > > > Regarding the communication bottleneck, we are not very sure what the reviewer means (we also asked the reviewer but didn’t hear back from the reviewer). We think the reviewer might be asking that as $N$ gets bigger, the simulation generates more data, and there might be a bottleneck in data transfer between the GPUs. This might be true in the sense that if we have a very big $N$ and when the GPU data transfer bandwidth is limited, we might get into a bottleneck in transferring data between the GPUs. However, in all our state-based experiments, we did not notice this being an issue. Moreover, the purpose of having $\beta_{a:v}, \beta_{p:v}$ is that we can control the ratios of the data generation, value, and policy updates. If one of them is slowed down, the others will wait to satisfy $\beta_{a:v}, \beta_{p:v}$ so that the learning is not adversely hurt. For example, if, for some reason, the v-learner did not get the data from the Actor for some time and keeps updating on the old rollout data, the $Q$ function might overfit the existing data, leading to bad policies. However, since we are explicitly controlling $\beta_{a:v}$, the value function updates will wait until new rollout data comes in. So explicitly controlling the $\beta_{p:v}, \beta_{a:v}$ can help reduce the variance in the performance on different hardware situations.
> > > >
> > > > > How is the batch size applied ? Is it applied solely on the Q and policy networks, or is it also applied to the actors ?
> > > >
> > > > We are sorry for the confusion here. The same batch size is applied to both the $Q$ function and policy function updates. For the Actor process (data collection process), there is no use of the batch size.
> > > >
> > > > > why is f_a defined as the number of rollouts per environment per unit of time instead of the total number of rollouts per unit of time?
> > > >
> > > > We kindly point out that the statement made by the reviewer, “The later could make the setting more robust to changes in the number of environments operating in parallel, and could have decreases the number of hyper-parameters.”, does not hold true. This statement is built upon the assumption that *if we generate the same amount of data regardless of the number of environments, then we can use the same $\beta_{a:v}$ ratio, meaning we can update the critic function the same amount of times*. However, empirically, we don’t find this is the case.
> > > >
> > > > Let’s see Fig 6 and take an example of the Shadow Hand task. Let’s denote the number of environments as $N$. To distinguish the reviewer’s suggested definition of $f_a^\prime$ and our definition of $f_a$, we denote the new ratio between the Actor process and the V-learner process using the reviewer’s definition of $f_a$ as $\beta^\prime_{a:v}$.  When $N=2048$, and $\beta_{a:v}=1:1$, which is equivalent to $\beta^\prime_{a:v}=2048:1$. Now let’s see when $N=8192$. According to what the reviewer suggests, we should see the performance of $(N=8192, \beta_{a:v}^\prime=8192:4,\beta_{a:v}=1:4)$ same as $(N=2048, \beta_{a:v}^\prime=2048:1,\beta_{a:v}=1:1)$. But as Fig 6 shows, these two scenarios lead to very different performance. Similarly, for the case of $(N=16384, \beta_{a:v}^\prime=16384:4,\beta_{a:v}=1:4)$ and $(N=4096, \beta_{a:v}^\prime=4096:1,\beta_{a:v}=1:1)$, there is a gap in the converged performance.
> > > >
> > > > These imply that the performance is still dependent on the number of environments. One explanation is that: rolling out $N$ environments for $T$ steps and rolling out $xN$ environments for $T/x$ steps give different exploration data. Consider an extreme case that $(N=4096,f_{a}=1, f_{a}^\prime=4096)$ and $(N=1, f_{a}=4096, f_{a}^\prime=4096)$, with the same amount of data collected, the performance could be highly different. So even if we change the definition of $f_a$, we will have the same amount of hyperparameters (we still need to tune $N$). Moreover, $f_a$ and $f_a^\prime$ are interchangeable and injective. Each $f_a$ can be uniquely mapped to $f_a^\prime$. So we don’t think using the new definition $f^\prime_a$ would make a big difference.

---

> > > > > ### Author Response · Authors · 2022-11-19
> > > > > **Response to Reviewer bLp3 (5/5)**
> > > > >
> > > > > > how are the values of a_u and a_l determined and how would the noise injection process change if the actions were discrete instead of continuous ?
> > > > >
> > > > > $a_u=1.0,~a_l=-1.0$ in all our experiments. As done in many continuous control environments (such as Mujoco continuous benchmark tasks, the Isaac Gym tasks, [https://github.com/NVIDIA-Omniverse/IsaacGymEnvs/blob/main/isaacgymenvs/tasks/base/vec_task.py#L105](https://github.com/NVIDIA-Omniverse/IsaacGymEnvs/blob/main/isaacgymenvs/tasks/base/vec_task.py#L105)), a common convention is to set the action space range to be $[-1,1]$. This is also why the policy network has a tanh activation function at the end in SAC. If the action space range of a continuous control task is not $[-1, 1]$, it is easy to scale the action space range to be $[-1,1]$ via a simple linear transformation ([https://github.com/openai/gym/blob/master/gym/wrappers/rescale_action.py](https://github.com/openai/gym/blob/master/gym/wrappers/rescale_action.py)).
> > > > >
> > > > > Our current PQL implementation focuses on continuous control tasks. However, it should be straightforward to modify it a bit to accommodate discrete control tasks. In discrete control tasks, one can use $\epsilon-$greedy exploration (no need for using Gaussian noise for exploration) as commonly done in DQN. And it is simple to apply the idea of mixed exploration here: just use different values of $\epsilon$ in different environments.

---

### Official Review · Reviewer_xfGC · 2022-10-26

**Confidence:** 2
**Correctness:** 3
**Technical Novelty And Significance:** 2
**Empirical Novelty And Significance:** 2
**Recommendation:** 5

**Clarity, Quality, Novelty And Reproducibility:**

The writing is easy to follow and clearly conveys the intended motivation and goals of the authors. The quality of the empirical data is good (though I found it a bit difficult to read some of the legends in Figures 3, 4, 7, 8, 9, 10). The novelty lies in the focus on training an off-policy algorithm with a large number of environments vs the typical setup with an on-policy algorithm and a relatively small number of environments (unless you have access to a large distributed system). I believe the results should be reproducible given access to the implementation.

**Strength And Weaknesses:**

Strengths
--------------------
- The fundamental question being addressed by the paper is interesting. The training dynamics of RL algorithms using GPU-simulated environments may be dramatically different from those of CPU-simulated environments. The abundance of environments producing data at a rate much higher than previously studied coupled with lower communication overheads, because data is generated directly on the device, provide ample motivation to rethink training algorithms and hyperparameters typically used by members of the community. In short, is PPO the best off-the-shelf algorithm in light of the differences between training environments?
- The authors demonstrate faster convergence for their offline training method compared to PPO on several GPU-simulated environments.
- The influence of several hyperparameters, such as batch-size, replay-buffer size, the number of training environments, and the number of GPUs were studied as part of the empirical results.

Weaknesses
-------------------
- The paper is empirically heavy but theoretically lite. There's no motivation underlying the choices for exploration of the design space of the parallel training environment hyperparameters.

**Summary Of The Paper:**

In this paper, the authors investigate the interplay between the training speed/performance of an offline RL method and the total number of environments used during training. A large number of RL papers default to online RL methods, specifically PPO, because of the relative ease of these methods, however, offline methods should be more sample efficient by virtue of reusing samples from the replay buffer. The authors implement an offline RL training scheme, PQL, to leverage recent advances in GPU simulation environments to accelerate the training process while achieving comparable results with online methods. PQL breaks up the data collection and training pipeline into 3 processes, an Actor, P-learner, and V-learner. Since the simulation environment is located on the GPU all 3 processes may be co-resident on a single GPU or distributed across multiple GPUs on a single node. The placement of processes on different GPUs and the ratio of data collection vs training speeds are studied empirically and optimal settings tend to vary, as expected, across environments. The authors also study empirically how the batch-size and replay-buffer sizes influence the training process as the number of processes changes.

**Summary Of The Review:**

For researchers that do not have access to a large pool of computational resources the ability to take maximum advantage of the resources they do have on hand is imperative. As such, on-policy methods typically underutilize GPU resources because they cannot generate a sufficient amount of data to keep the GPU saturated. Though many works have tried to address this issue by increasing the number of environments this typically requires an increase in the number of CPUs as well. I think it is quite interesting to try taking full advantage of the GPU-centric training environments to better utilize the hardware and investigate the interplay between the number of environments and the training dynamics. This type of investigation seems to be inline with the spirit and direction of the Issac Gym GPU simulator project.

The improved training time to accuracy is visible in a number of the plots. In Section 2 the authors reference an improvement in regard to both the sample efficiency and training times of off-policy algorithms but when I think of sample efficiency I would like to see plots of the return vs the number of environment interactions. The authors noted in section 4.6 that the volume of data generated by the simulators is so high that data in the replay buffers are being replaced rapidly causing the method to become more on-policy. I assume this means the sample efficiency also resembles that of an online policy in some cases, is that correct? Does this training scenario detract from one of the selling points of offline algorithms to reduce the volume of required training data?

In the discussion section the authors mention not using many modifications that are typical of modern offline algorithms. Was it too difficult to add these features on top of the current implementation? I feel that omitting these features could be a mistake since top-performing offline algorithms use them so without those reference points it's more challenging to compare PQL performance against prior work.

A concern is that the primary contribution is the framework that implements PQL, however, this artifact mostly builds on existing work (Issac Gym, PyTorch, etc). This diminishes the novelty of the work and makes the results appear more informative rather than a novel contribution that rises to the level required for publication at the current venue.

---

> ### Author Response · Authors · 2022-11-09
> **Request for clarification**
>
> Thank you for your comment and suggestions. While we are preparing the response, we are not very sure what you mean by "the choices for exploration of the design space of the parallel training environment hyperparameters." Could you please elaborate a bit more regarding this point so that we can better understand the question and explain it more clearly? Thanks.

---

> ### Author Response · Authors · 2022-11-19
> **Response to Reviewer xfGC (1/2)**
>
> We sincerely thank reviewer xfGC for the feedback on our paper. We address the reviewer's concerns as follows. There might be a typo in the reviewer’s Summary section. We would like to kindly point out that our work aims to improve the learning speed of off-policy methods. Investigation on the off-line RL methods is beyond the scope of this work.
>
> > The paper is empirically heavy but theoretically lite. There's no motivation underlying the choices for exploration of the design space of the parallel training environment hyperparameters.
>
> We are sorry if our motivation was unclear. The primary motivation underlying our design choices was to speed up training, improve performance and ablate the need for each hyperparameter. We have elaborated the motivation for hyperparameters here and modified the paper to make it clearer (shown in blue text).
>
> * **Number of Environments**: GPU simulation allows for running thousands of environments in parallel on a single workstation. We anticipate that GPU simulation is only going to improve with time. However, more parallel environments will only be useful if RL algorithms are able to exploit such data — in other words only if performance scales with more data. We, therefore, investigated how different algorithms scale with the number of environments, as shown in Figure 4.
> * **Batch size**: With many parallel environments ($N$), a lot of data is quickly generated. While one can easily increase $N$ from 100s to 10,000s in Isaac Gym on a single GPU, it is infeasible to increase the replay buffer size by 100 times due to the limited GPU memory or CPU RAM (if the data is stored on the CPU). Consequently, the replay buffer is overwritten frequently — meaning each collected sample may not be used efficiently. One way to efficiently utilize large amounts of changing data is to increase the batch size. To test how much increase in batch size is necessary for Q-learning with a limited capacity replay buffer to take advantage of the large amounts of incoming data, we investigated the relationship between performance and batch size.
> * **The effect of $\beta_{p:v}$ and $\beta_{a:v}$**: If $\beta_{p:v}$ is bigger, the policy updates more frequently than the value functions, potentially leading to policy overfitting to the stale value function, leading to bad exploration. If the policy updates much slower than the value function, then the policy might lag behind the value function a lot, which hurts the learning speed. Similarly, if $\beta_{a:v}$ is bigger, the V-learner might need to wait for the Actor to collect enough data as the simulation speed cannot be changed, leading to slower learning. If $\beta_{a:v}$ is smaller, the value function updates more given the generated rollout data.
>
> > I assume this means the sample efficiency also resembles that of an online policy in some cases, is that correct? Does this training scenario detract from one of the selling points of offline algorithms to reduce the volume of required training data?
>
> First, we would like to point out that we have provided the learning curves for sample efficiency in Fig. B.2 in the supplementary materials. As shown in Fig. B.2, our method PQL and DDPG consistently achieves better sample efficiency than PPO on all six tasks. On more challenging tasks such as Allegro Hand and Shadow Hand, the improvements in sample efficiency are even more salient. The SAC’s performance varies a bit: better sample efficiency than PPO on three tasks, and similar or worse sample efficiency on the other three tasks.
>
> Second, one of our contributions is speeding up the off-policy RL agent learning speed in terms of the wall clock time using massively parallel simulation, which comes at the cost of worse sample efficiency. In many practical applications, such as robotics, reducing the wall clock time is of greater interest than improving the sample efficiency. Consequently, our hyperparameters are designed and optimized to speed up the training wall clock time. For example, if one only uses 1 environment for data collection (which is a common practice in off-policy RL agent training), it’s expected that it will get a much better sample efficiency than if one uses 1000 parallel environments at the same time. But using more parallel environments greatly speeds up learning (also shown in Fig. 8).

---

> > ### Author Response · Authors · 2022-11-19
> > **Response to Reviewer xfGC (2/2)**
> >
> > > In the discussion section the authors mention not using many modifications that are typical of modern offline algorithms. Was it too difficult to add these features on top of the current implementation? I feel that omitting these features could be a mistake since top-performing offline algorithms use them so without those reference points it's more challenging to compare PQL performance against prior work.
> >
> > The implementation of additional techniques, such as prioritized experience replay (PER), is complementary to our work.
> > * First, it is unclear whether those techniques will be useful for sure in massively parallel simulation. The benefits of the additional techniques, such as PER, tend to vary across the tasks. In this work, we try to use fewer knobs while still achieving a good learning speed, which can simplify the implementation as well as keep the entire framework simple and more accessible. Our framework already achieves faster learning than the most commonly used RL algorithm (PPO) in Isaac Gym.
> > * Second, there is also a practical challenge in implementing PER. Given tens of thousands of parallel environments, how can one efficiently insert a big chuck of data into PER and update the sample priorities? Without solving this issue, it would cause a slowdown in training as it would consume a lot of time just to update the PER. We leave this investigation to future work.
> >
> > > A concern is that the primary contribution is the framework that implements PQL, however, this artifact mostly builds on existing work (Issac Gym, PyTorch, etc). This diminishes the novelty of the work and makes the results appear more informative rather than a novel contribution that rises to the level required for publication at the current venue.
> >
> > We use Isaac Gym as a simulator and Pytorch as the deep learning library. These are the tools that we use. Given Isaac Gym and Pytorch, almost all prior works (see Introduction and Related Work) use PPO for policy learning. It is not straightforward and trivial to come up with a better scheme that can learn faster than PPO. We wanna emphasize that our contribution is on how to design a fast RL learning framework on a single workstation using these tools. We achieve this by innovating how to parallelize off-policy algorithms and make them work with tens of thousands of parallel environments (orders of magnitude more environments than many prior distributed off-policy works).

---

> > > ### Comment · Reviewer_xfGC · 2022-12-09
> > > **Author response**
> > >
> > > First, I would like the thank the authors for their thorough attention to detail regarding the comments made during the initial review and provided ample clarification regarding my confusion regarding certain aspects of the presentation. I would also like to thank the authors for updating their submission to reflect the comments made by myself and other reviewers. I believe the work presented would be an interesting contribution to the RL community and help other researchers understand strategies to improve the utilization of their computational resources. However, after reviewing the concerns raised by the other reviewers I have, unfortunately, decided to lower my overall rating. I won't bother to reiterate those concerns here but I think as an RL systems-oriented paper it would be both motivational and persuasive to provide data regarding the GPU utilization as the number of actors, p-learners, and v-learners are varied. Maintaining high utilization throughout training seems to be an underlying principle that aids in training acceleration.

---

> > > > ### Author Response · Authors · 2022-12-09
> > > > **We kindly request the reviewer to elaborate on the reasons for lowering the score**
> > > >
> > > > We thank the reviewer for replying to our response. We are glad that the reviewer finds our paper to be making an interesting contribution to the RL community. We are sorry to see that the reviewer lowered the score. We request the reviewer to elaborate on the concerns that the reviewer has and leads to the lower score. If the reviewer is referring to the concerns raised by other reviewers, we believe that we have provided detailed explanations or experiments that helped address the concerns. If there is any **_specific_** remaining concern that bothers the reviewer, we kindly ask the reviewer to enlighten us, and we are more than happy to answer them.
> > > >
> > > > Regarding the concern about GPU utilization, as we mentioned in the paper, most of our experiments run on two GPUs. Our goal is to speed up the policy training given the compute resource. So we are less concerned about the GPU utilization percentage. The only thing we care about here is the amount of training wall-clock time for getting a good policy. And we agree that the GPU utilization is high because, with our distributed framework, each process runs continuously and each GPU gets maximum utilization. We are not sure why this is an issue. Just as an analogy, in supervised learning, if we increase the batch size for training the Imagenet classification network, of course, the GPU utilization will go up. But we don't consider this an issue as long as it leads to faster convergence. We wonder why this leads to a lower score. We would appreciate the reviewer elaborating on the concerns, and we are happy to provide more experimental results or more clarifications.

---

### Author Response · Authors · 2022-11-19
**General Response**

First, we would like to thank all the reviewers for their feedback and constructive suggestions on our manuscript. We are glad that the reviewers find our paper to be:

(1) addressing an interesting and highly-relevant problem (Reviewer xfGC, Reviewer bLp3, Reviewer BMNj, Reviewer iU6x)

(2) well-written and well-motivated (Reviewer xfGC, Reviewer BMNj), novel and non-trivial in the framework (Reviewer bLp3, Reviewer BMNj), intuitive and easy to follow (Reviewer BMNj)

(3) having extensive, comprehensive, informative, and good-quality experiments (Reviewer xfGC, Reviewer BMNj, Reviewer iU6x)

(4) providing enough details and reproducible (Reviewer BMNj, Reviewer iU6x)

We have also modified the writing and added more intuition behind each design choice (shown in blue text). We also explain our framework in the form of pseudo-code in the Appendix A. Our figures have been regenerated with a larger font size as well. Next, we will address each reviewer’s concerns individually.

---

### Author Response · Authors · 2022-12-13
**Summary of our rebuttal and discussion**

We sincerely thank all reviewers and ACs for their efforts and time in reviewing our paper and their constructive suggestions that strengthen our work.

To summarize our response:

**Contribution**:

We would like to emphasize again that our primary contributions are:
* We scale up off-policy methods such as DDPG to _tens of thousands of_ parallel environments.
* We propose a simple distributed training framework for a workstation setup by parallelizing the data collection, policy learning, and value function learning. We show that it substantially speeds up policy learning, achieving **new state-of-the-art** learning speeds on many massively parallel training environments.
* We demonstrate that off-policy methods can not only obtain better sample efficiency than on-policy methods such as PPO, but also achieves faster learning in terms of wall-clock time.
* We investigate what hyperparameters are essential to scale up off-policy methods and provide empirical tuning guidance.

**Additional Experiments**

* We have added experiments on a distributed version of SAC, and show that our distributed framework can indeed work effectively for other off-policy methods such as SAC.
* We have also compared our baseline implementations (PPO and SAC) with the widely-used RL codebase (RL-games) for Isaac Gym tasks and show that our implementations provide comparable performance.
* We also tried to compare our PQL with IMPALA (an on-policy distributed training framework). However, we didn't find it worked well on tasks with continuous action space. In fact, even maintainers of RLlib do not recommend using IMPALA for environments with continuous actions.

**Writing**

* We have added more intuition/motivation on how we developed our method, why our method works, and how we designed the experiments.
* We have increased the font size of all the figures and made sure they are clearly legible.
* We have added the pseudo-code for PQL in Appendix A.
* We have also addressed each reviewer's concern individually in the response below.

**Code Release**

* We promise to release our entire codebase, which will provide reliable and clean implementations of our method PQL, and all the baselines (DDPG, SAC with n-step returns, PPO). We believe PQL can be a new default algorithm that people use for learning policies in Isaac Gym.

We sincerely thank the reviewers for their insightful suggestions, which help improve our paper a lot. The additional experiments and writing modifications will be reflected in the final version of our paper as well.

If you have any remaining questions or concerns, please do not hesitate to let us know, we are more than happy to address them. We sincerely hope that we can resolve any misunderstanding or concern from the reviewers.

Sincerely,
Authors

---

### Decision · Program_Chairs · 2023-01-20

**Decision:**

Reject

**Justification For Why Not Higher Score:**

Would like to see a more systematic approach around the design and evaluation of the system, including design principles, alternatives considered, crucial assumptions, and robustness of findings to violations of the assumptions.

**Justification For Why Not Lower Score:**

N/A

**Metareview: Summary, Strengths And Weaknesses:**

This was a borderline paper. On one hand, reviewers appreciate the value of investigating the computational regime in RL where a GPU-based simulator is available, and results showing that a Q-learning based approach is competitive with PPO are interesting. On the other hand, reviewers were generally unsatisfied with the framing and presentation of the paper. While improvements made by the author response were appreciated, the main requests from reviewers were still for a more systematic approach to the systems considerations of the paper: what were the motivating design principles and theoretical basis for the implementation? More details about why the system is more efficient; What findings from these results are robust and transferable? A secondary concern was around the framing of the contribution, where reviewers found there to be less originality in the framework itself as compared to the implementation and systems considerations.

I'll also note that authors were unhappy with the level of engagement from the reviewers, and I agree it is unfortunate. However, all reviewers attended and participated in the in-person virtual discussion, and there was a relatively clear consensus amongst the reviewers coming out of the discussion.

**Summary Of Ac-Reviewer Meeting:**

There was surprisingly little disagreement amongst reviewers. The most positive reviewer appreciated that there were lots of experiments but was reconsidering after reading other reviews and author responses, and they struggled to articulate the value of the contributions when challenged by other reviewers. There was some discussion questioning the novelty of the contributions around the framework. At one point, one of the reviewers proposed viewing the paper through the lens of a systems paper and pointed the group to usenix.org/conferences/author-resources/how-and-how-not-write-good-systems-paper, raising questions like mentioned in the "Lessons", "Choices", and "Context" sections there. This resonated with the other reviewers, even after looking at author responses.